# POLICY LEARNING USING WEAK SUPERVISION

## ABSTRACT

Most existing policy learning solutions require the learning agents to receive high-quality supervision signals, e.g., rewards in reinforcement learning (RL) or high-quality expert's demonstrations in behavioral cloning (BC). These quality supervisions are either infeasible or prohibitively expensive to obtain in practice. We aim for a unified framework that leverages the weak supervisions to perform policy learning efficiently. To handle this problem, we treat the "weak supervisions" as imperfect information coming from a *peer agent*, and evaluate the learning agent's policy based on a "correlated agreement" with the peer agent's policy (instead of simple agreements). Our way of leveraging peer agent's information offers us a family of solutions that learn effectively from weak supervisions with theoretical guarantees. Extensive evaluations on tasks including RL with noisy reward, BC with weak demonstrations and standard policy co-training (RL + BC) show that the proposed approach leads to substantial improvements, especially when the complexity or the noise of the learning environments grows.

## 1 INTRODUCTION

Recent breakthrough in policy learning (PL) opens up the possibility to apply these techniques in real-world applications such as robotics (Mnih et al., 2015; Akkaya et al., 2019) and self-driving (Bojarski et al., 2016a; Codevilla et al., 2018). Nonetheless, most existing works require agents to receive high-quality supervision signals, e.g., reward or expert's demonstrations, which are either infeasible or prohibitively expensive to obtain in practice. For instance, (1) the reward may be collected through sensors thus not credible (Everitt et al., 2017; Romoff et al., 2018; Wang et al., 2020); (2) the demonstrations by an expert in behavioral cloning (BC) are often imperfect due to limited resources and environment noise (Laskey et al., 2017; Wu et al., 2019; Reddy et al., 2020).

Learning from weak supervision signals such as noisy rewards $\tilde{r}$ (noisy versions of $r$) (Wang et al., 2020) or low-quality demonstrations $\widetilde{\mathcal{D}}_E$ (noisy versions of $\mathcal{D}_E$) produced by problematic expert $\tilde{\pi}_E$ (Wu et al., 2019) is one of the outstanding challenges that prevents a wider application of PL. Although some recent works have explored these topics separately in their specific domains (Guo et al., 2019; Wang et al., 2020; Lee et al., 2020), there is no unified solution towards performing robust policy learning pip install arxiv-latex-cleaner under this imperfect supervision. In this work, we first formulate a meta-framework to study RL/BC with weak supervision signals and call it *weakly supervised policy learning*. Then as a response, we propose a theoretically principled solution concept, PeerPL, to perform efficient policy learning using the available weak supervisions.

Our solution concept is inspired by the literature of peer prediction (Miller et al., 2005; Dasgupta & Ghosh, 2013; Shnayder et al., 2016), where the question concerns verifying information without ground truth verification. Instead, a group of agents' reports (none of which is assumed to be high-quality nor clean) are used to validate each other's information. We adopt a similar idea and treat the "weak supervisions" as information coming from a *peer agent*, and evaluate the learning agent's policy based on a "correlated agreement" (CA) with the peer agent's. Compared to standard reward/loss functions that impose simple agreements with the weak supervisions, our approach punishes an over-agreement to avoid overfitting to the weak supervisions. Our way of leveraging peer agent's information offers us a family of solutions that 1) does not require prior knowledge of the weakness of the supervisions, and 2) learns effectively with strong theoretical guarantees.

We demonstrate how the proposed PeerPL framework adapts in challenging tasks including RL with noisy rewards and behavioral cloning (BC) from weak demonstrations. Furthermore, we provide

intensive analysis of the convergence behavior and the sample complexity for our solutions. These results jointly demonstrate that our approach enables agents to learn the optimal policy efficiently under weak supervisions. Evaluations on these tasks show strong evidence that PeerPL brings significant improvements over state-of-the-art solutions, especially when the complexity or the noise of the learning environments grows.

To summarize, the contributions in the paper are mainly three-folds: (1) We provide a unified formulation of *weakly supervised policy learning* to model the weak supervision in RL/BC problems; (2) We propose a novel PeerPL solution framework based on calculating a correlated agreement with weak supervisions, a novel way for policy evaluation introduced to RL/BC tasks; (3) PeerPL is theoretically guaranteed to recover the optimal policy (as if the supervisions are of high-quality and clean) and competitive empirical performances are observed in several policy learning tasks.

## 2 RELATED WORK

**Learning with Noisy Supervision**   Learning from noisy labels is widely explored within the supervised learning domain. Beginning from the seminal work (Natarajan et al., 2013) that first proposed an unbiased surrogate loss function to recover the true loss given the knowledge of noise rates, follow-up works focus on how to estimate the noise rates based on noisy observations (Scott et al., 2013; Scott, 2015; Sukhbaatar & Fergus, 2014; van Rooyen & Williamson, 2015; Menon et al., 2015). Recent work (Wang et al., 2020) adapts this idea within RL and proposes a statistics-based estimation algorithm. However, the estimation is not efficient especially when the state-action space is huge. Moreover, as a sequential process, the error in estimating the noise rate can accumulate and amplify when deploying an RL algorithm. In contrast, our solution in this paper does not require a priori specification of the noise rates thus offloading the burden of estimation.

**Behavioral Cloning (BC)**   Standard BC (Pomerleau, 1991; Ross & Bagnell, 2010) tackles the sequential decision-making problem by imitating the expert's actions using supervised learning. Specifically, it aims to minimize the one-step deviation error over the expert's trajectory without reasoning the sequential consequences of actions. Therefore, the agent suffers from compounding errors when there is a mismatch between demonstrations and real states encountered (Ross & Bagnell, 2010; Ross et al., 2011). Recent works introduce data augmentations (Bojarski et al., 2016b) and value-based regularization (Reddy et al., 2019) or inverse dynamics models (Torabi et al., 2018; Monteiro et al., 2020) to encourage learning long-horizon behaviors. While simple and straightforward, BC has been widely investigated in a wide range of domains (Giusti et al., 2016; Justesen & Risi, 2017) and often yields competitive performance (Farag & Saleh, 2018; Reddy et al., 2019). Our framework is complementary to current BC literature by introducing a learning strategy from weak demonstrations (e.g., noisy or from a poorly-trained agent) and provides theoretical guarantees on how to retrieve clean policy under mild assumptions (Song et al., 2019).

**Correlated Agreement**   Peer prediction aims to elicit information from self-interested agents without ground-truth verification (Miller et al., 2005; Dasgupta & Ghosh, 2013; Shnayder et al., 2016). The only source of information to serve as verification is from the agents' reports. Particularly, in Dasgupta & Ghosh (2013); Shnayder et al. (2016), a correlated agreement (CA) type of mechanism is proposed, which evaluates the correlations between agents' reports. In addition to encouraging some agreement between agents, CA mechanism also punishes over-agreement when two agents always report identically. This property helps reduce the effect of noisy reports by punishing overfitting to them. Recently, Liu & Guo (2020) adapts a similar idea in learning from noisy labels for supervised learning. We consider a more challenging weakly supervised policy learning setting and study the convergence rates in sequential decision-making problems.

## 3 POLICY LEARNING FROM WEAK SUPERVISION

We begin by introducing a general framework to unify PL with low-quality supervision signals. Then we provide instantiations of the proposed weakly supervised formulation with two different applications: (1) RL with noisy reward and (2) behavioral cloning (BC) using weak expert demonstrations.

### 3.1 PRELIMINARY OF POLICY LEARNING

The goal of policy learning (PL) is to learn a policy $\pi$ that the agent could follow to perform a series of actions in a stateful environment. For RL, the interactive environment is characterized as an MDP $\mathcal{M} = \langle \mathcal{S}, \mathcal{A}, \mathcal{R}, \mathcal{P}, \gamma \rangle$. At each time $t$, the agent in state $s_t \in \mathcal{S}$ takes an action $a_t \in \mathcal{A}$ by following the policy $\pi : \mathcal{S} \times \mathcal{A} \to \mathbb{R}$, and *potentially* receives a reward $r(s_t, a_t) \in \mathcal{R}$. Then the agent transfers to the next state $s_{t+1}$ according to a transition probability function $\mathcal{P}$. We denote the generated trajectory $\tau = \{(s_t, a_t, r_t)\}_{t=0}^T$, where $T$ is a finite or infinite horizon. RL algorithms aim to maximize the expected reward over the trajectory $\tau$ induced by the policy: $\underline{J(\pi) = \mathbb{E}_{(s_t, a_t, r_t) \sim \tau}[\sum_{t=0}^T \gamma^t r_t]}$, where $\gamma \in (0, 1]$ is the discount factor.

Another popular policy learning method is through behavioral cloning (BC). The goal of BC is to mimic the expert policy $\pi_E$ through demonstrations $D_E = \{(s_i, a_i)\}_{i=1}^N$ drawn from distribution $\mathcal{D}_E$ (generated according to $\pi_E$), where $(s_i, a_i)$ is the sampled state-action pair from the expert's trajectory. Typically, training a policy with standard BC corresponds to maximizing the following log-likelihood: $\underline{J(\pi) = \mathbb{E}_{(s,a) \sim \mathcal{D}_E}[\log \pi(a|s)]}$.

In both RL and BC, the agent receives "supervisions" through either the reward $r$ by interacting with environments or the expert policy $\pi_E$ as observable demonstrations. Consider a particular policy class $\Pi$, the optimal policy is then defined as $\pi^* = \arg\max_{\pi \in \Pi} J(\pi)$: $\pi^*$ obtains maximum expected reward over the horizon $T$ in RL and $\underline{\pi^* \text{ corresponds to the clean expert policy } \pi_E \text{ in BC}}$. In practice, one can also leverage both RL and BC approaches to take advantage of both worlds (Brys et al., 2015; Hester et al., 2018; Guo et al., 2019; Song et al., 2019). Specifically, a recent hybrid framework called policy co-training (Song et al., 2019) is considered in this paper.

### 3.2 META FRAMEWORK FOR POLICY LEARNING WITH WEAK SUPERVISION

To unify, we denote these *weak supervision* signals using $\widetilde{Y}$ that is either the reward $\tilde{r}$ for RL or the action $\tilde{a}$ performed by an expert policy $\tilde{\pi}_E$ for BC. $\widetilde{Y}$ denotes a weak version of a high-quality supervision signal $Y$. As a consequence, in an abstract manner, a weakly supervised PL problem can be formulated as learning the optimal policy $\pi^*$ with only accessing a weak supervision sequence denoting as $\{(s_i, a_i), \widetilde{Y}_i\}_{i=1}^N$.

To unify the discussion, suppose that we have an evaluation function $\mathsf{Eva}_\pi((s_i, a_i), \widetilde{Y}_i)$ that evaluates a taken policy at state $(s_i, a_i)$ with a weak supervision $\widetilde{Y}_i$. In the RL setting, this $\mathsf{Eva}_\pi$ is the loss for different RL algorithms, which is a function of noisy reward $\tilde{r}$ received at $(s_i, a_i)$. While for the BC setting, this $\mathsf{Eva}_\pi$ is the loss used to evaluate the action taken by the agent given the action taken by the expert. Furthermore, we let $J(\pi)$ denote the function that evaluates policy $\pi$ under a set of state action pairs with weak supervision signals $\{(s_i, a_i), \widetilde{Y}_i\}_{i=1}^N$, i.e., $J(\pi) = \mathbb{E}_{(s,a) \sim \tau}[\mathsf{Eva}_\pi((s, a), \tilde{Y})]$. Note that above unified notations are only for better delivery of our framework and we still treat PL as a sequential decision problem. We focus on the following two instantiations for our weakly supervised settings:

**RL with Noisy Reward** Consider a finite MDP $\widetilde{\mathcal{M}} = \langle \mathcal{S}, \mathcal{A}, \mathcal{R}, F, \mathcal{P}, \gamma \rangle$ with noisy reward channels (Wang et al., 2020), where $\mathcal{R} : \mathcal{S} \times \mathcal{A} \to \mathbb{R}$, and the noisy reward $\tilde{r}$ is generated following a certain function $F : \mathcal{R} \to \widetilde{\mathcal{R}}$. Denote the trajectory a policy $\pi_\theta$ generates via interacting with $\widetilde{\mathcal{M}}$ as $\tilde{\tau}_\theta$. Assume the reward is discrete and has $|\mathcal{R}|$ levels, the noise rate can be characterized via a matrix $\mathbf{C}_{|\mathcal{R}| \times |\mathcal{R}|}^{\mathrm{RL}}$, where each entry $c_{j,k}$ indicates the flipping probability for generating a perturbed outcome: $c_{j,k}^{\mathrm{RL}} = \mathbb{P}(\tilde{r}_t = R_k | r_t = R_j)$. We call $r$ and $\tilde{r}$ the *true reward* and *noisy reward* respectively.

**BC with Weak Demonstration** Instead of observing the expert demonstration generated according to $\pi_E$, denote the available weak demonstrations by $\{(s_i, \tilde{a}_i)\}_{i=1}^N$, where $\tilde{a}_i \sim \tilde{\pi}_E(\cdot|s_i)$ is the noisy action and each state-action pair $(s_i, \tilde{a}_i)$ is drawn from distribution $\widetilde{\mathcal{D}}_E$. In particular, we assume the noisy action $\tilde{a}_i$ is independent of the state $s$ given the deterministic expert action $\pi_E(s)$, i.e., $\mathbb{P}(\tilde{a}_i | \pi_E(s_i)) = \mathbb{P}(\tilde{a}_i | s_i, \pi_E(s_i))$. Similar to RL, we assume the noise regime can be characterized by a confusion matrix $\mathbf{C}_{|\mathcal{A}| \times |\mathcal{A}|}^{\mathrm{BC}}$, where each entry $c_{j,k}$ indicates the flipping probability for the expert to take an suboptimal action $c_{j,k}^{\mathrm{BC}} = \mathbb{P}(\tilde{\pi}_E(s_k) = A_k | \pi_E(s_j) = A_j)$. In this setting, we'd like to recover $\pi^*$ as if we were able to access the quality expert demonstration $\pi_E$ instead of $\tilde{\pi}_E$.

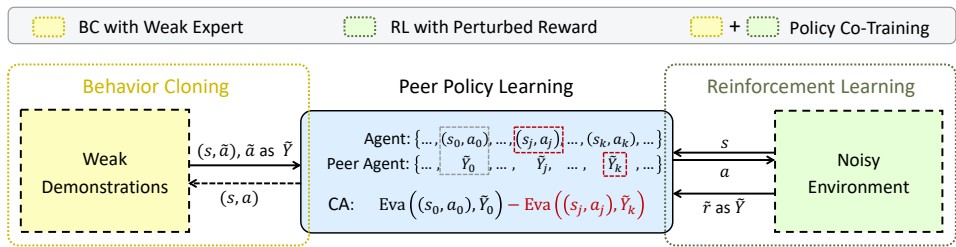

Figure 1: Illustration of *weakly supervised policy learning* and our PeerPL solution with correlated agreement (CA). We use $\tilde{Y}$ to denote a weak supervision, be it a noisy reward, or a noisy demonstration. Eva stands for an evaluation function. "Peer Agent" corresponds to weak supervisions.

**Remark** We emphasize that we do not have nor need the knowledge of the weakness of the signals, i.e., $\mathbf{C}^{\mathrm{RL}}_{|\mathcal{R}| \times |\mathcal{R}|}$ nor $\mathbf{C}^{\mathrm{BC}}_{|\mathcal{A}| \times |\mathcal{A}|}$. As we show later, our approach successfully avoided the need of this knowledge. This is also a main challenge for designing our solution.

## 4 PEERPL: WEAKLY SUPERVISED PL VIA CORRELATED AGREEMENT

To deal with weak supervisions in PL, we propose a unified and theoretically principled framework PeerPL. We treat the weak supervisions as information coming from a "peer agent", and then evaluates the policy using a certain type of "correlated agreement" function between the learning policy and the peer agent.

### 4.1 OVERVIEW OF THE IDEA: CORRELATED AGREEMENT WITH WEAK SUPERVISIONS

We first present the general idea of our PeerPL framework that uses a concept named "correlated agreement" (CA). For each weakly supervised state-action pair $((s_i, a_i), \widetilde{Y}_i)$, we randomly sample a state-action pair $(s_j, a_j), j \neq i$, as well as another supervision signal $\widetilde{Y}_k, k \neq i, j$ from a different state-action pair. Then we evaluate $((s_i, a_i), \widetilde{Y}_i)$ according to the following:

$$\text{CA with Weak Supervision: } \mathsf{Eva}_\pi\big((s_i, a_i), \widetilde{Y}_i\big) - \mathsf{Eva}_\pi\big((s_j, a_j), \widetilde{Y}_k\big) \tag{1}$$

Intuitively, the first term above encourages an "agreement" with the weak supervision (that a policy agrees with the corresponding supervision), while the second term punishes a "blind" and "over" agreement that happens when the agent's policy always matches with the weak supervision even on randomly paired traces (noise). The randomly paired samples $j, k$ helps us achieve this check. Note the implementation of our mechanism does not require the knowledge of $\mathbf{C}^{\mathrm{RL}}_{|\mathcal{R}| \times |\mathcal{R}|}$ nor $\mathbf{C}^{\mathrm{BC}}_{|\mathcal{A}| \times |\mathcal{A}|}$, and offers a **prior-knowledge free** way to learn effectively with weak supervsions. This above process is illustrated in Figure 1.

**Illustrative toy example** Consider a toy BC setting where we learned a policy that outputs a sequence of actions $a_1 = a_2 = a_3 = 1, a_4 = 0$, and they have perfectly matched the weak supervision $a'_1 = a'_2 = a'_3 = 1, a'_4 = 0$ (at the same sequence of states). Supposedly $\mathsf{Eva}_\pi((s_i, a_i), a'_i)$ evaluates how well a policy matches the expert demonstration ($\mathsf{Eva}_\pi = 1$ for agreeing, 0 for not). Using only $\mathsf{Eva}_\pi((s_i, a_i), a'_i)$ will return the highest score 1 for agreeing with this noisy/imperfect/low-quality supervision. However the correlated agreement evaluation returns (for example for $i = 1$) $\mathbb{E}[\mathsf{Eva}_\pi((s_i, a_i), a'_i) - \mathsf{Eva}_\pi((s_j, a_j), a'_k, k \neq j)] = 1 - (0.75^2 + 0.25^2) = 0.375$, where $0.75^2 + 0.25^2$ is the probability of randomly paired $a$ and $a'$ match each other. The above example shows that a full agreement with the weak supervision will instead be punished!

In what follows, we solidate our implementations within each of the settings considered and provide theoretical guarantees under weak supervision.

### 4.2 PEERRL: PEER REINFORCEMENT LEARNING

Since the reward signals are no longer credible in the weakly supervised RL setting, we propose the following objective function that punishes the over-agreement based on CA mechanism

$$J^{\mathrm{RL}}(\pi_\theta) = \mathbb{E}\Big[\mathsf{Eva}^{\mathrm{RL}}_\pi\big((s_i, a_i), \tilde{r}_i\big)\Big] - \xi \cdot \mathbb{E}\Big[\mathsf{Eva}^{\mathrm{RL}}_\pi\big((s_j, a_j), \tilde{r}_k\big)\Big], \tag{2}$$

$$\text{where} \quad \mathsf{Eva}^{\mathrm{RL}}_\pi\big((s, a), \tilde{r}\big) = -\ell\big(\pi_\theta, (s, a, \tilde{r})\big). \tag{3}$$

In (2), the first expectation is taken over $(s_i, a_i, \tilde{r}_i) \sim \tilde{\tau}_\theta$ and second one is taken over $(s_j, a_j, \tilde{r}_j) \sim \tilde{\tau}_\theta, (s_k, a_k, \tilde{r}_k) \sim \tilde{\tau}_\theta$, where $\tilde{\tau}_\theta$ is the trajectory generated by $\pi_\theta$ with the noisy reward function $\tilde{r}$. The choice of the loss function $\ell$ depends on the RL algorithms used (e.g., temporal difference error (Mnih et al., 2013; Wang et al., 2016) or the policy gradient loss (Sutton et al., 1999)). Also, the learning sequence is encoded in $\pi$, therefore, maximizing the objective $J^{\mathrm{RL}}(\pi)$ is equivalent to maximizing the accumulated reward. $\xi \geq 0$ is a hyperparameter to balance the penalty for blind agreements induced by CA.

In what follows, we consider the $Q$-Learning (Watkins & Dayan, 1992) as the underlying learning algorithm where $\ell(\pi_\theta, (s, a, \tilde{r})) = -\tilde{r}$ and demonstrate our CA mechanism provides strong guarantees for $Q$-Learning with only observing the noisy reward. For clarity, we define *peer reward* as

$$\text{Peer RL Reward:} \quad \tilde{r}_{\mathrm{peer}}(s, a) = \tilde{r}(s, a) - \xi \cdot \tilde{r}',$$

where $r'$ is a randomly sampled reward over all state-action pair and $\xi \geq 0$ is a parameter to balance the noisy reward and the punishment for blind agreement (with $\tilde{r}'$). We set $\xi = 1$ (for binary case) in the following analysis and treat each $(s, a)$ equally when sampling the $r'$. In our experiment, $\tilde{r}_{\mathrm{peer}}$ is not sensitive to the choice of $\xi$, and we have kept $\xi$ a constant for each run of the RL experiments.

We show peer reward $\tilde{r}_{\mathrm{peer}}$ offers us an affine transformation of the true reward in expectation, which is the key to guaranteeing the convergence of our Peer RL algorithm to converge to $\pi^*$. For clarity, consider the binary reward setting ($r_+$ and $r_-$) and denote the error in $\tilde{r}$ as $e_+ = \mathbb{P}(\tilde{r} = r_- | r = r_+), e_- = \mathbb{P}(\tilde{r} = r_+ | r = r_-)$ (a simplification of $\mathbf{C}_{|\mathcal{R}| \times |\mathcal{R}|}^{\mathrm{RL}}$ in the binary setting).

**Lemma 1.** *Let $r \in [0, R_{\max}]$ be a bounded reward. Assume $1 - e_- - e_+ > 0$, then we have:*

$$\mathbb{E}[\tilde{r}_{\mathrm{peer}}] = (1 - e_- - e_+) \cdot \mathbb{E}[r_{\mathrm{peer}}] = (1 - e_- - e_+) \cdot \mathbb{E}[r] + const,$$

*where $r_{peer} = r_{peer}(s, a) = r(s, a) - \xi \cdot r'$ is the peer RL reward when observing the true reward $r$.*

Lemma 1 shows that by subtracting the peer penalty term $\tilde{r}'$ from noisy reward $\tilde{r}$, $\tilde{r}_{\mathrm{peer}}$ recovers the clean and true reward $r$ in expectation.

**Remark** It's notable that the expectation of the noisy reward $\mathbb{E}[\tilde{r}]$ can also be written as:

$$\mathbb{E}[\tilde{r}] = (1 - e_- - e_+)\mathbb{E}[r] + \underbrace{e_- r_+ + e_+ r_-}_{const}.$$

However, we claim that the constant in peer reward has less effect on the true reward $r$. That's because we have that

$$\mathbb{E}[\tilde{r}] = (1 - e_- - e_+) \cdot \left( \mathbb{E}[r] + \frac{e_+}{1 - e_- - e_+} r_- + \frac{e_-}{1 - e_- - e_+} r_+ \right),$$

$$\mathbb{E}[\tilde{r}_{\mathrm{peer}}] = (1 - e_- - e_+) \cdot (\mathbb{E}[r] - (1 - p_{\mathrm{peer}}) r_- - p_{\mathrm{peer}} r_+),$$

where $p_{\mathrm{peer}} \in [0, 1]$ denotes the probability that a sample policy gets a reward $r_+$. Since the magnitude of noise terms $\frac{e_-}{1 - e_- - e_+}$ and $\frac{e_+}{1 - e_- - e_+}$ can potentially become much larger than $1 - p_{\mathrm{peer}}$ and $p_{\mathrm{peer}}$ in a high-noise regime, $\frac{e_-}{1 - e_- - e_+} r_+ + \frac{e_+}{1 - e_- - e_+} r_-$ will dilute the informativeness of $\mathbb{E}[r]$. On the contrary, $\mathbb{E}[\tilde{r}_{\mathrm{peer}}]$ contains a moderate constant noise thus maintaining more useful training signals of the true reward in practice.

Based on Lemma 1, we further offer the following convergence guarantee:

**Theorem 1.** *(Convergence) Given a finite MDP with noisy reward, denoting as $\widetilde{\mathcal{M}} = \langle \mathcal{S}, \mathcal{A}, \mathcal{R}, F, \mathcal{P}, \gamma \rangle$, the $Q$-learning algorithm with peer rewards, given by the update rule,*

$$Q_{t+1}(s_t, a_t) = (1 - \alpha_t) Q_t(s_t, a_t) + \alpha_t [\tilde{r}_{\mathrm{peer}}(s_t, a_t) + \gamma \max_{a' \in \mathcal{A}} Q_t(s_{t+1}, a')], \pi_t(s) = \arg\max_{a \in \mathcal{A}} Q_t(s, a)$$

*converges w.p.1 to the optimal policy $\pi^*(s)$ as long as $\sum_t \alpha_t = \infty$ and $\sum_t \alpha_t^2 < \infty$.*

Theorem 1 states that the agent will converge to the optimal policy *w.p.1* with peer rewards without requiring any knowledge of the corruption in rewards ($\mathbf{C}_{|\mathcal{R}| \times |\mathcal{R}|}^{\mathrm{RL}}$, as opposed to previous work (Wang et al., 2020) that requires such knowledge). Moreover, we found that, to guarantee the convergence to $\pi^*$, the number of samples needed for our approach is no more than $\mathcal{O}(1/(1 - e_- - e_+)^2)$ times of the one needed when the RL agent observes true rewards perfectly (see Appendix A).

**Remark** Even though we only presented analysis for the binary case for $Q$-Learning, our approach is rather generic and is ready to be plugged into modern DRL algorithms. Specifically, we provide the **multi-reward extension** and implementations with DQN (Mnih et al., 2013) and policy gradient (Sutton et al., 1999) using peer reward in Appendix A.

### 4.3 PEERBC: PEER BEHAVIORAL CLONING

Similarly, we present our CA solution in the setting of behavioral cloning (PeerBC). In BC, the supervision is given by the weak expert' noisy trajectory. The $\mathsf{Eva}_\pi^{\mathrm{BC}}$ function in BC evaluates the agent policy $\pi_\theta$ and the weak expert' trajectory $\{(s_i, \tilde{a}_i)\}_{i=1}^N$ using $\ell(\pi_\theta, (s_i, \tilde{a}_i))$ where $\ell$ is an arbitrary classification loss. Taking for instance the cross-entropy, the objective of PeerBC is:

$$J^{\mathrm{BC}}(\pi_\theta) = \mathbb{E}\Big[\mathsf{Eva}_\pi^{\mathrm{BC}}\big((s_i, a_i), \tilde{a}_i\big)\Big] - \xi \cdot \mathbb{E}\Big[\mathsf{Eva}_\pi^{\mathrm{BC}}\big((s_j, a_j), \tilde{a}_k\big)\Big], \tag{4}$$

$$\text{where} \quad \mathsf{Eva}_\pi^{\mathrm{BC}}\big((s, a), \tilde{a}\big) = -\ell\big(\pi_\theta, (s, \tilde{a})\big) = \log \pi_\theta(\tilde{a}|s). \tag{5}$$

In (4), the first expectation is taken over $(s_i, \tilde{a}_i) \sim \widetilde{\mathcal{D}}_E, a_i \sim \pi(\cdot|s_i)$ and the second is taken over $(s_j, \tilde{a}_j) \sim \widetilde{\mathcal{D}}_E, a_j \sim \pi(\cdot|s_j), (s_k, \tilde{a}_k) \sim \widetilde{\mathcal{D}}_E, a_k \sim \pi(\cdot|s_k)$. Again, the second $\mathsf{Eva}_\pi^{\mathrm{BC}}$ term in $J^{\mathrm{BC}}$ serves the purpose of punishing over-agreement with the weak demonstration. Similarly, $\xi \geq 0$ is a parameter to balance the penalty for blind agreements. At each iteration, the agent learns under weak supervision $\tilde{a}$, and the training samples are generated from the distribution $\widetilde{\mathcal{D}}_E$ determined by the weak expert.

We prove that the policy learned by PeerBC converges to the expert policy when observing a sufficient amount of weak demonstrations. We focus on the binary action setting for the purpose of theoretical analysis, where the action space is given by $\mathcal{A} = \{A_+, A_-\}$ and the weakness or noise in the weak expert $\tilde{\pi}_E$ is quantified by $e_+ = \mathbb{P}(\tilde{\pi}_E(s) = A_-|\pi_E(s) = A_+)$ and $e_- = \mathbb{P}(\tilde{\pi}_E(s) = A_+|\pi_E(s) = A_-)$ (a simplification of $\mathbf{C}_{|\mathcal{A}|\times|\mathcal{A}|}^{\mathrm{BC}}$ in the binary setting). Let the $\pi_{\widetilde{\mathcal{D}}_E}$ be the optimal policy by maximizing the objective in (4) with imperfect demonstrations $\widetilde{\mathcal{D}}_E$ (a particular set of with $N$ i.i.d. imperfect demonstrations). Note $\ell(\cdot)$ should be specified as indicator loss: $\mathbb{1}(\pi(s), a) = 1$ when $\pi(s) \neq a$, otherwise $\mathbb{1}(\pi(s), a) = 0$. We have the following upper bound on the error rate.

**Theorem 2.** *Denote by* $R_{\widetilde{\mathcal{D}}_E} := \mathbb{P}_{(s,a)\sim\mathcal{D}_E}(\pi_{\widetilde{\mathcal{D}}_E}(s) \neq a)$ *the error rate for PeerBC. With probability at least* $1 - \delta$, *it is upper-bounded as:* $R_{\widetilde{\mathcal{D}}_E} \leq \frac{1+\xi}{1-e_- -e_+} \sqrt{\frac{2\log 2/\delta}{N}}$.

Theorem 2 states that as long as weak demonstrations are observed sufficiently, i.e., $N$ is sufficiently large, the policy learned by PeerBC is able to converge to the clean expert policy $\pi_E(s)$ with a convergence rate of $\mathcal{O}(1/\sqrt{N})$.

**Peer Policy Co-Training** Our discussion of BC allows us to study a more challenging co-training task (Song et al., 2019). Given a finite MDP $\mathcal{M}$, there are two agents that receive partial observations and we let $\pi^A$ and $\pi^B$ denote the policies for agent $A$ and $B$. Moreover, two agents are trained jointly to learn with rewards and noisy demonstrations from each other (e.g., at preliminary training phase). Symmetrically, we consider on the case where agent $A$ learns with the demonstrations from $B$ on sampled trajectories, and $\pi_B$ effectively serves as a noisy version of expert policy $\pi_E = \arg\max_{\pi\in\Pi} \mathbb{E}_{(s,a)\sim\mathcal{D}_E}[\log \pi(a|s)]$.

---

**Algorithm 1** Peer policy co-training (PeerCT)

**Require:** Views $A$, $B$, MDPs $\mathcal{M}^A$, $\mathcal{M}^B$, policies $\pi_A, \pi_B$, mapping functions $f_{A\rightarrow B}, f_{B\rightarrow A}$ that maps states from one view to the other view, CA coefficient $\xi$, step size $\beta$ for policy update.

1: **repeat**
2:      Run $\pi^A$ to generate trajectories $\tau^A = \{(s_i^A, a_i^A, r_i^A)\}_{i=1}^N$.
3:      Run $\pi^B$ to generate trajectories $\tau^B = \{(s_j^B, a_j^B, r_j^B)\}_{j=1}^M$.
4:      Agents label the trajectories for each other

$$\tau'^A \leftarrow \left\{\big(s_i^A, \pi^B\big(f_{B\leftarrow A}(s_i^A)\big)\big)\right\}_{i=1}^N,$$
$$\tau'^B \leftarrow \left\{\big(s_j^B, \pi^A\big(f_{A\leftarrow B}(s_j^B)\big)\big)\right\}_{j=1}^M.$$

5:      Update policies: $\pi^{\{A,B\}} \leftarrow \pi^{\{A,B\}} + \beta \cdot \nabla J^{\mathrm{CT}}(\pi^{\{A,B\}})$
6: **until** convergence

---

For simplicity of demonstration, we focus on recovering the clean expert policy by only adapting the BC evaluation term (ignoring the effect of RL rewards, see Eqn. (6)). Denote by $\tau_\theta^A = \{(s_i^A, a_i^A, r_i^A)\}_{i=1}^N$ the trajectory that $\pi^A$ generated via interacting with the partial world

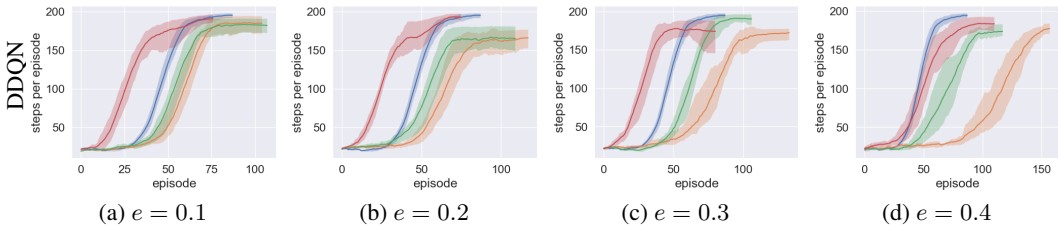

Figure 2: Learning curves of DDQN on CartPole with true reward ($r$) ■, noisy reward ($\tilde{r}$) ■, surrogate reward (Wang et al., 2020) ($\hat{r}$) ■, and peer reward ($\tilde{r}_{\text{peer}}$, $\xi = 0.2$) ■.

$\mathcal{M}^A$. Then $\pi^B$ substitutes each action $a_i^A$ with its selection $a_i'^B \sim \pi^B(\cdot | f_{A \to B}(s_i^A))$ as the weak supervisions. Similar to PeerRL/PeerBC, the objective function of peer co-learning (PeerCT) becomes

$$J^{\text{CT}}(\pi_\theta) = \mathbb{E}\Big[\mathsf{Eva}_\pi^{\text{RL}}\big((s_i^A, a_i^A), r_i^A\big) + \mathsf{Eva}_\pi^{\text{BC}}\big((s_i^A, a_i^A), a_i'^B\big)\Big] - \xi \cdot \mathbb{E}\Big[\mathsf{Eva}_\pi^{\text{BC}}\big((s_j^A, a_j^A), a_k'^B\big)\Big], \quad (6)$$

where the first expectation is taken over $(s_i^A, a_i^A, r_i^A) \sim \tau_\theta^A$, $a_i'^B \sim \pi^B(\cdot | f_{A \to B}(s_i^A))$, and the second is taken over $(s_j^A, a_j^A, r_j^A) \sim \tau_\theta^A$, $(s_k^A, a_k^A, r_k^A) \sim \tau_\theta^A$, $a_k'^B \sim \pi^B(\cdot | f_{A \to B}(s_k^A))$, $\ell$ is the loss function defined in Eqn. (5) to measure the policy difference, and $\mathsf{Eva}_\pi^{\text{RL}}$, $\mathsf{Eva}_\pi^{\text{BC}}$ are defined in Eqn. (3) and (5) respectively. The full algorithm PeerCT is provided in Algorithm 1. We omit detailed discussions on the convergence of PeerCT due to it can be viewed as a straight-forward extension of Theorem 2 in the context of co-training.

## 5 EXPERIMENTS

We evaluate our solution in three challenging weakly supervised PL problems. Experiments on control games and Atari show that, without any prior knowledge of the noise in supervisions, our approach is able to leverage weak supervisions more effectively.

### 5.1 PEERRL WITH NOISY REWARD

We first evaluate our method in RL with noisy reward setting. Following Wang et al. (2020), we consider the binary reward $\{-1, 1\}$ for Cartpole where the symmetric noise is synthesized with different error rates $e = e_- = e_+$. We choose DQN (Mnih et al., 2013) and Dueling DQN (DDQN) (Wang et al., 2016) algorithms and train the models for 10,000 steps. We repeat each experiment 10 times with different random seeds and leave the results for DQN in Appendix D.

Figure 2 shows the learning curves for DDQN with different approaches in noisy environments ($\xi = 0.2$)[1]. Since the number of training steps is fixed, the faster the algorithm converges, the fewer total episodes the agent will involve thus the learning curve is on the left side. As a consequence, the proposed peer reward outperforms other baselines significantly even in a high-noise regime (e.g., $e = 0.4$). Moreover, we highlight that peer reward does not require any knowledge of noise rates or complicated estimation algorithms compared to Wang et al. (2020). Table 1 provides quantitative results on the average reward $\mathcal{R}_{avg}$ and total episodes $N_{epi}$. We find the agents with peer reward lead to a larger $\mathcal{R}_{avg}$ (less generalization error) and a smaller $N_{epi}$ (faster convergence) consistently, which again verifies the effectiveness of our approach.

**Analysis of the benefits in PeerRL** When the noise rate $e$ is small, we observed that the agents with peer reward even lead to faster convergence than ones observing true reward perfectly. This indicates there might be other factors apart from noise reduction that promote the RL learning with peer reward. We hypothesize this is because (1) peer reward scales the reward signals appropriately, which potentially reduces the variance and makes it easier to learn from; (2) the peer penalty term encourages explorations in RL implicitly; (3) the human-specific "true reward" is also imperfect which leads to a weak supervision scenario. More discussions and analysis are deferred to Appendix C.6. There might be multiple explanations for the improvement in performance provided by PeerRL depending on the specific RL tasks. However, we emphasize that the property of recovering from noise is non-negligible especially in a high-noise regime (e.g., $e = 0.4$).

---

[1]We analysed the sensitivity of $\xi$ and found the algorithm performs reasonable when $\xi \in (0.1, 0.4)$. More insights and experiments with varied $\xi$ is deferred to Appendix D.

Table 1: Numerical performance of DDQN on CartPole with true reward ($r$), noisy reward ($\tilde{r}$), surrogate reward $\hat{r}$ (Wang et al., 2020), and peer reward $\tilde{r}_{\text{peer}}(\xi = 0.2)$. $\mathcal{R}_{avg}$ denotes average reward per episode after convergence, the higher ($\uparrow$) the better; $N_{epi}$ denotes total episodes involved in 10,000 steps, the lower ($\downarrow$) the better.

|  |  | $e = 0.1$ | | $e = 0.2$ | | $e = 0.3$ | | $e = 0.4$ | |
|---|---|---|---|---|---|---|---|---|---|
|  |  | $\mathcal{R}_{avg}\uparrow$ | $N_{epi}\downarrow$ | $\mathcal{R}_{avg}\uparrow$ | $N_{epi}\downarrow$ | $\mathcal{R}_{avg}\uparrow$ | $N_{epi}\downarrow$ | $\mathcal{R}_{avg}\uparrow$ | $N_{epi}\downarrow$ |
| DDQN | $r$ | $195.6 \pm 3.1$ | $101.2 \pm 3.2$ | $195.6 \pm 3.1$ | $101.2 \pm 3.2$ | $195.6 \pm 3.1$ | $101.2 \pm 3.2$ | $195.2 \pm 3.0$ | $101.2 \pm 3.3$ |
|  | $\tilde{r}$ | $185.2 \pm 15.6$ | $114.6 \pm 6.0$ | $168.8 \pm 13.6$ | $123.9 \pm 9.6$ | $177.1 \pm 11.2$ | $133.2 \pm 9.1$ | $185.5 \pm 10.9$ | $163.1 \pm 11.0$ |
|  | $\hat{r}$ | $183.9 \pm 10.4$ | $110.6 \pm 6.7$ | $165.1 \pm 18.2$ | $113.9 \pm 9.6$ | $\mathbf{192.2 \pm 10.9}$ | $115.5 \pm 4.3$ | $179.2 \pm 6.6$ | $125.8 \pm 9.6$ |
|  | $\tilde{r}_{\text{peer}}$ | $\mathbf{198.5 \pm 2.3}$ | $\mathbf{86.2 \pm 5.0}$ | $\mathbf{195.5 \pm 9.1}$ | $\mathbf{85.3 \pm 5.4}$ | $174.1 \pm 32.5$ | $\mathbf{88.8 \pm 6.3}$ | $\mathbf{191.8 \pm 8.5}$ | $106.9 \pm 9.2$ |

| (a) Pong | (b) Boxing | (c) Enduro | (d) Freeway |
|---|---|---|---|

Figure 3: Learning curves of BC on Atari. Standard BC ■, PeerBC (ours) ■, expert ■.

## 5.2 PEERBC FROM WEAK DEMONSTRATIONS

In BC setting, we evaluate our approach on four vision-based Atari games. For each environment, we train an imperfect RL model with PPO (Schulman et al., 2017) algorithm. Here, "imperfect" means the training is terminated before convergence when the performance is about $70\% \sim 90\%$ as good as the fully

Table 2: BC from weak demonstrations. Our approach successfully recovers better policies than expert.

| Environment | | Pong | Boxing | Enduro | Freeway | Lift ($\uparrow$) |
|---|---|---|---|---|---|---|
| Expert | | $15.1 \pm 6.6$ | $67.5 \pm 8.5$ | $150.1 \pm 23.0$ | $21.9 \pm 1.7$ | - |
| Standard BC | | $14.7 \pm 3.2$ | $56.2 \pm 7.7$ | $138.9 \pm 14.1$ | $22.0 \pm 1.3$ | $-6.6\%$ |
| PeerBC | $\xi = 0.2$ | $\mathbf{18.8 \pm 0.6}$ | $67.2 \pm 8.4$ | $177.9 \pm 29.3$ | $\mathbf{22.5 \pm 0.6}$ | $+11.3\%$ |
|  | $\xi = 0.5$ | $16.6 \pm 4.0$ | $\mathbf{75.6 \pm 5.4}$ | $230.9 \pm 73.0$ | $22.4 \pm 1.3$ | $\mathbf{+19.5\%}$ |
|  | $\xi = 1.0$ | $16.7 \pm 4.3$ | $69.7 \pm 4.7$ | $230.4 \pm 61.6$ | $8.9 \pm 4.9$ | $+2.0\%$ |
| Fully converged PPO | | $20.9 \pm 0.3$ | $89.3 \pm 5.4$ | $389.6 \pm 216.9$ | $33.3 \pm 0.8$ | $+70.6\%$ |

converged model. We then collect the imperfect demonstrations using the expert model and generate 100 trajectories for each environment. The results are reported under three random seeds.

Figure 3 presents the comparisons for standard BC and PeerBC, from which we observe that our approach outperforms standard BC and even the expert it learns from. Note that during the whole training process, the agent never learns by interacting directly with the environment but only have access to the expert's trajectories. Therefore, we owe this performance gain to PeerBC's strong ability for learning from weak supervisions. The peer term we add not only provably eliminates the effects of noise but also extracts useful strategy from the demonstrations. In Table 2, we show the quantitative results. Our approach consistently outperforms the expert and standard BC. As a reference, we also compare two other baselines GAIL (Ho & Ermon, 2016) and SQIL (Reddy et al., 2019) and provide the sensitivity analysis of $\xi$ in Appendix D.

**Analysis of benefits in PeerBC** Similarly, the performance improvement of PeerBC might be also coupled with multiple possible factors. (1) The imperfect expert model might be a noisy version of the fully-converged agent since there are less visited states on which the selected actions of the model contains noise. (2) The improvements might be brought up by biasing against high-entropy policies thus PeerBC is useful when the true policy itself is deterministic. We provide more discussions about the second factor in Appendix C.5.

## 5.3 PEERCT FOR STANDARD POLICY CO-TRAINING

Finally, we verify the effectiveness of PeerCT algorithm in policy co-training setting (Song et al., 2019). This setting is more challenging since the states are partially observable and each agent needs to imitate another agent's behavior that is highly biased and imperfect. Note that we adopt the exact same setting as Song et al. (2019) **without any synthetic noise** included. This implies the potential of our approach to deal with natural noise in real-world applications. Following Song et al. (2019), we mask the first two dimensions respectively in the state vector to create two views for co-training in classic control games (Acrobot and CartPole). Similarly, the agent either removes all even index

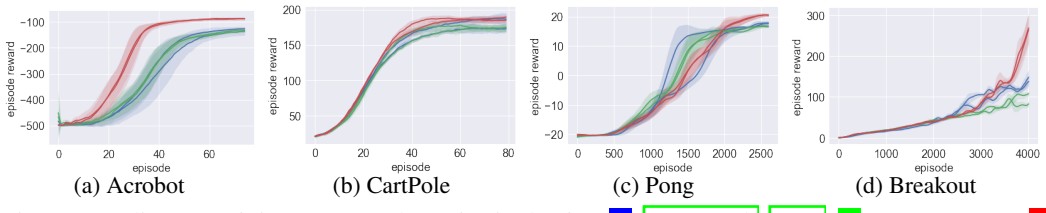

Figure 4: Policy co-training on control/Atari. Single view ▮, Song et al. (2019) ▮, PeerCT (ours) ▮.

coordinates (view-$A$) in the state vector or removing all odd index ones (view-$B$) on Atari games. As shown in Table 3 and Figure 4, PeerCT algorithm outperforms training from single view, and CoPiEr algorithm consistently on both control games ($\xi = 0.5$ in Figure 4a, 4b) and Atari games ($\xi = 0.2$ in Figure 4c, 4d). In most cases, our approach leads to a faster convergence and lower generalization error compared to CoPiEr, which again verify that our ways of leveraging information from peer agent enables recovery of useful knowledge from highly imperfect supervisions.

## 6 CONCLUSION

To deal with a series of RL/BC problems with low-quality supervision signals, we formulate a meta-framework *weakly supervised policy learning* to unify the problem instances with weak supervision in sequential decision-making. Inspired by the correlated agreement (CA) mechanism, we propose a theoretical

Table 3: Comparison with single view training and CoPiEr (Song et al., 2019) on standard policy co-training.

| Environment | | Acrobot | CartPole | Pong | Breakout | Lift ($\uparrow$) |
|---|---|---|---|---|---|---|
| Single View | A | $-136.6 \pm 15.6$ | $172.8 \pm 5.5$ | $17.8 \pm 0.6$ | $148.0 \pm 16.5$ | 34.7% |
| | B | $-126.4 \pm 8.0$ | $186.7 \pm 8.1$ | $17.7 \pm 0.5$ | $137.8 \pm 12.5$ | 35.5% |
| CoPiEr | A | $-136.2 \pm 5.2$ | $174.1 \pm 5.1$ | $16.8 \pm 0.5$ | $107.5 \pm 5.8$ | 52.9% |
| | B | $-131.5 \pm 4.5$ | $174.3 \pm 5.4$ | $16.5 \pm 0.2$ | $82.7 \pm 6.9$ | **72.0**% |
| PeerCT | A | $\mathbf{-87.0 \pm 3.9}$ | $\mathbf{188.8 \pm 2.7}$ | $\mathbf{20.5 \pm 0.4}$ | $263.6 \pm 36.0$ | - |
| | B | $-87.1 \pm 6.3$ | $184.7 \pm 3.9$ | $20.4 \pm 0.5$ | $\mathbf{268.6 \pm 33.6}$ | - |

principled framework PeerPL that builds on evaluating a learning policy's correlated agreements with the weak supervisions. We demonstrate how our method adapts in RL/BC and the combined co-training tasks and provide intensive analyses of their convergence behaviors and sample complexity. Experiments on these tasks show our approach leads to substantial improvements over baseline methods.

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

# A ANALYSIS OF PEERRL

We start this section by providing the proof of the convergence of $Q$-Learning under peer reward $\tilde{r}_{\text{peer}}$ (Theorem 1). Moreover, we give the sample complexity of *phased* value iteration (Theorem A1). In the rest of this section, we show how to extend the proposed method to multi-outcome setting (Section A.3) and modern deep reinforcement learning (DRL) algorithms such as policy gradient (Sutton et al., 1999) and DQN (Mnih et al., 2013; van Hasselt et al., 2016) (Section A.4).

## A.1 CONVERGENCE

Recall that we consider the binary reward case $\{r_+, r_-\}$, where $r_+$ and $r_-$ are two reward levels. The flipping errors of the reward are defined as $e_+ = \mathbb{P}(\tilde{r}_t = r_- | r_t = r_+)$ and $e_- = \mathbb{P}(\tilde{r}_t = r_+ | r_t = r_-)$. The *peer reward* is defined as $r_{\text{peer}}(s, a) = r(s, a) - r'$, where $r'$ is randomly sampled reward over all state-action pair $(s, a)$. Note that we treat each $(s, a)$ equally when sampling the $r'$ due to lack of the knowledge of true transition probability $\mathcal{P}$. In practice, the agent could only noisy observation of peer reward $\tilde{r}_{\text{peer}}(s, a) = \tilde{r}(s, a) - \tilde{r}'$. We provide the $Q$-learning with peer reward in Algorithm A1.

---

**Algorithm A1** $Q$-Learning with Peer Reward

---

**Require:** $\widetilde{\mathcal{M}} = (\mathcal{S}, \mathcal{A}, \widetilde{\mathcal{R}}, \mathcal{P}, \gamma)$, learning rate $\alpha \in (0, 1)$, initial state distribution $\beta_0$.
1: Initialize $Q: \mathcal{S} \times \mathcal{A} \to \mathbb{R}$ arbitrarily
2: **while** $Q$ is not converged **do**
3:     Start in state $s \sim \beta_0$
4:     **while** $s$ is not terminal **do**
5:         Calculate $\pi$ according to $Q$ and exploration strategy
6:         $a \leftarrow \pi(s)$; $s' \sim \mathcal{P}(\cdot | s, a)$
7:         Observe noisy reward $\tilde{r}(s, a)$ and randomly sample another $\tilde{r}'$ from all state-action pairs
8:         Calculate peer reward $\tilde{r}_{\text{peer}}(s, a) = \tilde{r}(s, a) - \tilde{r}'$
9:         $Q(s, a) \leftarrow (1 - \alpha) \cdot Q(s, a) + \alpha \cdot (\tilde{r}_{\text{peer}}(s, a) + \gamma \cdot \max_{a'} Q(s', a'))$
10:         $s \leftarrow s'$
11:     **end while**
12: **end while**
13: **return** $Q(s, a)$ and $\pi(s)$

---

We then show the proposed peer reward $\tilde{r}_{\text{peer}}$ offers us an affine transformation of true reward in expectation, which is the key to guaranteeing the convergence for RL algorithms.

**Lemma 1.** *Let $r \in [0, R_{\max}]$ be bounded reward and assume $1 - e_- - e_+ > 0$. Then, if we define the peer reward $\tilde{r}_{\text{peer}}(s, a) = \tilde{r}(s, a) - \tilde{r}'$, in which the penalty term $\tilde{r}'$ is randomly sampled noisy reward over all state-action pair $(s, a)$, we have*

$$\mathbb{E}[\tilde{r}_{\text{peer}}] = (1 - e_- - e_+)\mathbb{E}[r_{\text{peer}}] = (1 - e_- - e_+)\mathbb{E}[r] + const,$$

*where $r_{peer}$ is the clean version of peer reward when observing the true reward.*

*Proof.* With slight notation abuse, we let $\tilde{r}_{\text{peer}}, r, \tilde{r}, r'$ also represent the random variables. Let $\pi$ denotes the RL agent's policy. Consider the two terms on the RHS of noisy peer reward separately,

$$\mathbb{E}[\tilde{r}] = \mathbb{P}(r = r_+ | \pi) \cdot \mathbb{E}_{r=r_+} [\mathbb{P}(\tilde{r} = r_- | r = r_+) \cdot r_- + \mathbb{P}(\tilde{r} = r_+ | r = r_+) \cdot r_+] \tag{7}$$

$$+ \mathbb{P}(r = r_- | \pi) \cdot \mathbb{E}_{r=r_-} [\mathbb{P}(\tilde{r} = r_- | r = r_-) \cdot r_- + \mathbb{P}(\tilde{r} = r_+ | r = r_-) \cdot r_+] \tag{8}$$

$$= \mathbb{P}(r = r_+ | \pi) \cdot \mathbb{E}_{r=r_+} [e_+ r_- + (1 - e_+) r_+] \tag{9}$$

$$+ \mathbb{P}(r = r_- | \pi) \cdot \mathbb{E}_{r=r_-} [(1 - e_-) r_- + e_- r_+] \tag{10}$$

$$= \mathbb{P}(r = r_+ | \pi) \cdot \mathbb{E}_{r=r_+} [(1 - e_+ - e_-) \cdot r_+ + e_+ r_- + e_- r_+] \tag{11}$$

$$+ \mathbb{P}(r = r_- | \pi) \cdot \mathbb{E}_{r=r_-} [(1 - e_- - e_+) \cdot r_- + e_- r_+ + e_+ r_-)] \tag{12}$$

$$= (1 - e_+ - e_-)\mathbb{E}[r] + e_- r_+ + e_+ r_-. \tag{13}$$

Since we are treating the visitation probability of all state-action pair $(s, a)$ equally while sampling the peer penalty $r'$, then the probability of true reward $r$ under this sampling policy $\pi_{\text{sample}}$ is a

constant, denoting as $p_{\text{peer}}$, i.e., $p_{\text{peer}} = \mathbb{P}(r = r_- | \pi_{\text{sample}})$ is a constant. Then we have,

$$\mathbb{E}[\tilde{r}'] = \mathbb{P}(\tilde{r} = r_- | \pi_{\text{sample}}) \cdot r_- + \mathbb{P}(\tilde{r} = r_+ | \pi_{\text{sample}}) \cdot r_+ \tag{14}$$

$$= (e_+ p_{\text{peer}} + (1 - e_-)(1 - p_{\text{peer}})) \cdot r_- + ((1 - e_+) p_{\text{peer}} + e_-(1 - p_{\text{peer}})) \cdot r_+ \tag{15}$$

$$= (1 - e_- - e_+)[(1 - p_{\text{peer}}) \cdot r_- + p_{\text{peer}} \cdot r_+] + e_+ r_- + e_- r_+. \tag{16}$$

As a consequence, we obtain the expectation of peer reward satisfies

$$\mathbb{E}[\tilde{r}_{\text{peer}}] = \mathbb{E}[\tilde{r}] - \mathbb{E}[\tilde{r}'] \tag{17}$$

$$= (1 - e_+ - e_-)\mathbb{E}[r] - (1 - e_- - e_+)[(1 - p_{\text{peer}}) \cdot r_- + p_{\text{peer}} \cdot r_+] \tag{18}$$

$$= (1 - e_- - e_+)\mathbb{E}[r] + \text{const}. \tag{19}$$

Similarly, it is easy to obtain that $\mathbb{E}[r_{\text{peer}}] = \mathbb{E}[r] - [(1 - p_{\text{peer}}) \cdot r_- + p_{\text{peer}} \cdot r_+]$. Therefore, we have $\mathbb{E}[\tilde{r}_{\text{peer}}] = (1 - e_- - e_+)\mathbb{E}[r_{\text{peer}}] = (1 - e_- - e_+)\mathbb{E}[r] + \text{const}.$ □

Lemma 1 shows the proposed peer reward $\tilde{r}_{\text{peer}}$ offers us a "noise-free" positive ($1 - e_- - e_+ > 0$) linear transformation of true reward $r$ in expectation, which is shown the key to govern the convergence. It is widely known in utility theory and reward shaping literature (Ng et al., 1999; Asmuth et al., 2008; Von Neumann & Morgenstern, 2007) that any positive linear transformations leave the optimal policy unchanged. As a consequence, we consider a "transformed MDP" $\hat{\mathcal{M}}$ with reward $\hat{r} = (1 - e_- - e_+)r + \text{const}$, where the const is the same as the constant in Eqn. (19).

In what follows, we provide the formulation of the concept of "transformed MDP" with the policy invariance guarantee.

**Lemma A1.** *Given a finite MDP $\mathcal{M} = \langle \mathcal{S}, \mathcal{A}, \mathcal{R}, \mathcal{P}, \gamma \rangle$, a transformed MDP $\hat{\mathcal{M}} = \langle \mathcal{S}, \mathcal{A}, \hat{\mathcal{R}}, \mathcal{P}, \gamma \rangle$ with positive linear transformation in reward $\hat{r} := a \cdot r + b$, where $a, b$ are constants and $a > 0$, is guaranteed consistency in optimal policy.*

*Proof.* The $Q$ function for transformed MDP $\hat{\mathcal{M}}$ (denoting as $\hat{Q}$) is given as follows:

$$\hat{Q}(s, a) = \sum_{t=0}^{\infty} \gamma^t \hat{r}_t = \sum_{t=0}^{\infty} \gamma^t (a \cdot r_t + b)$$

$$= a \sum_{t=0}^{\infty} \gamma^t r_t + \sum_{t=0}^{\infty} \gamma^t b$$

$$= a \cdot Q(s, a) + B,$$

where $B = \sum_{t=0}^{\infty} \gamma^t b$ is a constant. Therefore, there is only a postive linear shift ($a > 0$) in $\hat{Q}(s, a)$ thus resulting in invariance in optimal policy for transformed MDP:

$$\hat{\pi}^*(s) = \arg\max_{a \in \mathcal{A}} \hat{Q}^*(s, a) = \arg\max_{a \in \mathcal{A}} [a \cdot Q(s, a) + B]$$

$$= \arg\max_{a \in \mathcal{A}} Q(s, a) = \pi^*(s).$$

□

Lemma A1 states that we only need to analysis the convergence of learned policy $\pi(s)$ to the optimal policy $\hat{\pi}^*(s)$ for transformed MDP $\hat{\mathcal{M}}$, which is equivalent to the optimal policy $\pi(s)^*$ for original MDP. This result is relevant to potential-based reward shaping (Ng et al., 1999; Asmuth et al., 2008) where a specific class of state-dependent transformation is adopted to speed up the convergence speed of $Q$-Learning meanwhile maintaining the optimal policy invariance. Moreover, a degenerate case for single-step decisions is studied in utility theory (Von Neumann & Morgenstern, 2007) which also implies our result.

Finally, we need an auxiliary result (Lemma A2) from stochastic process approximation to analyse the convergence for $Q$-Learning.

**Lemma A2.** *The random process $\{\Delta_t\}$ taking values in $\mathbb{R}^n$ and defined as*

$$\Delta_{t+1}(x) = (1 - \alpha_t(x))\Delta_t(x) + \alpha_t(x)F_t(x)$$

*converges to zero w.p.1 under the following assumptions:*

- $0 \leq \alpha_t \leq 1$, $\sum_t \alpha_t(x) = \infty$ *and* $\sum_t \alpha_t(x)^2 < \infty$;

- $||\mathbb{E}\left[F_t(x)|\mathcal{F}_t\right]||_W \leq \gamma||\Delta_t||$, *with* $\gamma < 1$;

- ***Var*** $\left[F_t(x)|\mathcal{F}_t\right] \leq C(1 + ||\Delta_t||_W^2)$, *for* $C > 0$.

*Here* $\mathcal{F}_t = \{\Delta_t, \Delta_{t-1}, \cdots, F_{t-1} \cdots, \alpha_t, \cdots\}$ *stands for the past at step t, $\alpha_t(x)$ is allowed to depend on the past insofar as the above conditions remain valid. The notation $|| \cdot ||_W$ refers to some weighted maximum norm.*

*Proof of Lemma A2.* See previous literature (Jaakkola et al., 1993; Tsitsiklis, 1994). □

**Theorem 1.** *(Convergence) Given a finite MDP with noisy reward, denoting as $\widetilde{\mathcal{M}} = \langle \mathcal{S}, \mathcal{A}, \widetilde{\mathcal{R}}, F, \mathcal{P}, \gamma \rangle$, the Q-learning algorithm with peer rewards, given by the update rule,*

$$Q_{t+1}(s_t, a_t) = (1 - \alpha_t)Q_t(s_t, a_t) + \alpha_t \left[ \tilde{r}_{\text{peer}}(s_t, a_t) + \gamma \max_{b \in \mathcal{A}} Q_t(s_{t+1}, b) \right], \quad (20)$$

$$\pi_t(s) = \arg\max_{a \in \mathcal{A}} Q_t(s, a) \quad (21)$$

*converges w.p.1 to the optimal policy $\pi^*(s)$ as long as $\sum_t \alpha_t = \infty$ and $\sum_t \alpha_t^2 < \infty$.*

*Proof.* Firstly, we construct a surrogate MDP $\hat{\mathcal{M}}$ with the positive-linearly transformed reward $\hat{r} = (1 - e_- - e_+) \cdot r + \text{const}$, where $\text{const} = -(1 - e_- - e_+)((1 - p) \cdot r_- + p \cdot r_+)$ is a constant. From Lemma A1, we know the optimal policy for $\hat{\mathcal{M}}$ is precisely the optimal policy for $\mathcal{M}$: $\hat{\pi}^*(s) = \pi^*(s)$.

Let $\hat{Q}^*$ denotes the optimal state-action function for this transformed MDP $\hat{\mathcal{M}}$. For notation brevity, we abbreviate $s_t$, $s_{t+1}$, $\tilde{r}_{\text{peer}}(s_t, s_{t+1})$, $Q_t$, $Q_{t+1}$, and $\alpha_t$ as $s$, $s'$, $Q$, $Q'$, $\tilde{r}_{\text{peer}}$ and $\alpha$, respectively.

Subtracting from both sides the quantity $\hat{Q}^*(s, a)$ in Eqn. (21):

$$Q'(s,a) - \hat{Q}^*(s,a) = (1 - \alpha)\left(Q(s,a) - \hat{Q}^*(s,a)\right) + \alpha \left[ \tilde{r}_{\text{peer}} + \gamma \max_{b \in \mathcal{A}} Q(s', b) - \hat{Q}^*(s,a) \right].$$

Let $\Delta_t(s,a) = Q(s,a) - \hat{Q}^*(s,a)$ and $F_t(s,a) = \tilde{r}_{\text{peer}} + \gamma \max_{b \in \mathcal{A}} Q(s', b) - \hat{Q}^*(s,a)$.

$$\Delta_{t+1}(s',a) = (1 - \alpha)\Delta_t(s,a) + \alpha F_t(s,a).$$

In consequence,

$$\mathbb{E}\left[F_t(s,a)|\mathcal{F}_t\right] = \mathbb{E}\left[\tilde{r}_{\text{peer}} + \gamma \max_{b \in \mathcal{A}} Q(s', b)\right] - \hat{Q}^*(s,a)$$

$$= \mathbb{E}\left[\tilde{r}_{\text{peer}} + \gamma \max_{b \in \mathcal{A}} Q(s', b) - \hat{r} - \gamma \max_{b \in \mathcal{A}} \hat{Q}^*(s', b)\right]$$

$$= \mathbb{E}\left[\tilde{r}_{\text{peer}}\right] - \mathbb{E}\left[\hat{r}\right] + \gamma\mathbb{E}\left[\max_{b \in \mathcal{A}} Q(s', b) - \max_{b \in \mathcal{A}} \hat{Q}^*(s', b)\right]$$

$$= \gamma\mathbb{E}\left[\max_{b \in \mathcal{A}} Q(s', b) - \max_{b \in \mathcal{A}} \hat{Q}^*(s', b)\right]$$

$$\leq \gamma\mathbb{E}\left[\max_{b \in \mathcal{A}, s' \in \mathcal{S}} \left|Q(s', b) - \hat{Q}^*(s', b)\right|\right]$$

$$= \gamma\mathbb{E}\left[||Q - \hat{Q}^*||_\infty\right] = \gamma||Q - \hat{Q}^*||_\infty = \gamma||\Delta_t||_\infty.$$

In above derivations, we utilize the unbiasedness property for peer reward (Lemma 1) and the inequality $\max_{b \in \mathcal{A}} Q(s', b) - \max_{b \in \mathcal{A}} \hat{Q}^*(s', b) \leq \max_{b \in \mathcal{A}, s' \in \mathcal{S}} \left| Q(s', b) - \hat{Q}^*(s', b) \right|$.

$$\mathbf{Var}\left[F_t(s,a)|\mathcal{F}_t\right] = \mathbb{E}\left[\left(\left(\tilde{r}_{\text{peer}} + \gamma \max_{b \in \mathcal{A}} Q(s', b) - \hat{Q}^*(s,a) - \mathbb{E}\left[\tilde{r}_{\text{peer}} + \gamma \max_{b \in \mathcal{A}} Q(s', b) - \hat{Q}^*(s,a)\right]\right)\right)^2\right]$$

$$= \mathbb{E}\left[\left(\left(\tilde{r}_{\text{peer}} + \gamma \max_{b \in \mathcal{A}} Q(s', b) - \mathbb{E}\left[\tilde{r}_{\text{peer}} + \gamma \max_{b \in \mathcal{A}} Q(s', b)\right]\right)\right)^2\right]$$

$$= \mathbf{Var}\left[\tilde{r}_{\text{peer}} + \gamma \max_{b \in \mathcal{A}} Q(s', b)\right].$$

Since $\tilde{r}_{\text{peer}}$ is bounded, it can be clearly verified that

$$\mathbf{Var}\left[F_t(s,a)|\mathcal{F}_t\right] \leq C''(1 + ||\Delta_t(s,a)||_\infty^2)$$

for some constant $C'' > 0$. Then, $\Delta_t$ converges to zero w.p.1 from Lemma A2, *i.e.*, $Q(s,a)$ converges to $\hat{Q}^*(s,a)$. As a consequence, we know the policy $\pi_t(s)$ converges to the optimal policy $\hat{\pi}^*(s) = \pi^*(s)$. □

### A.2 SAMPLE COMPLEXITY

In this section, we establish the sample complexity for $Q$-Learning with peer reward as discussed in Sec 4.2. Since the transition probability $\mathcal{P}$ in MDP remains unknown in practice, we firstly introduce a practical sampling model $G(\mathcal{M})$ following previous literature (Kearns & Singh, 1998; 2000; Kearns et al., 1999). in which the transition can be observed by calling the generative model. Then the sample complexity is analogous to the number of calls for $G(\mathcal{M})$ to obtain a near optimal policy.

**Definition A1.** *A generative model $G(\mathcal{M})$ for an MDP $\mathcal{M}$ is a sampling model which takes a state-action pair $(s_t, a_t)$ as input, and outputs the corresponding reward $r(s_t, a_t)$ and the next state $s_{t+1}$ randomly with the probability of $\mathbb{P}_a(s_t, s_{t+1})$, i.e., $s_{t+1} \sim \mathbb{P}(\cdot|s, a)$.*

It is known that exact value iteration is not feasible when the agent interacts with generative model $G(\mathcal{M})$ (Wang et al., 2020; Kakade, 2003). For the convenience of analysing sample complexity, we introduce a *phased value iteration* following Wang et al. (2020); Kearns & Singh (1998); Kakade (2003).

---

**Algorithm A2** Phased Value Iteration

---

**Require:** $G(\mathcal{M})$: generative model of $\mathcal{M} = (\mathcal{S}, \mathcal{A}, \mathcal{R}, \mathcal{P}, \gamma)$, $T$: number of iterations.
1: Set $V_T(s) = 0$
2: **for** $t = T - 1, \cdots, 0$ **do**
3:     Calling $G(\mathcal{M})$ $m$ times for each state-action pair.

$$\bar{\mathbb{P}}_a(s_t, s_{t+1}) = \frac{\#[(s_t, a_t) \to s_{t+1}]}{m}$$

4:     Set

$$V(s_t) = \max_{a \in \mathcal{A}} \sum_{s_{t+1} \in \mathcal{S}} \bar{\mathbb{P}}_a(s_t, s_{t+1}) \left[r_t + \gamma V(s_{t+1})\right]$$

$$\pi(s) = \arg\max_{a \in \mathcal{A}} V(s_t)$$

5: **end for**
6: **return** $V(s)$ and $\pi(s)$

---

Note that $\bar{P}_a(s_t, s_{t+1})$ is the estimation of transition probability $P_a(s_t, s_{t+1})$ by calling $G(\mathcal{M})$ m times. For the simplicity of notations, the iteration index $t$ decreases from $T - 1$ to 0.

We could also adopt peer reward in phased value iteration by replacing Line 4 in Algorithm A2 by

$$V(s_t) = \max_{a \in \mathcal{A}} \sum_{s_{t+1} \in \mathcal{S}} \bar{\mathbb{P}}_a(s_t, s_{t+1}) \left[\tilde{r}_{\text{peer}}(s_t, a) + \gamma V(s_{t+1})\right].$$

Then the sample complexity of one variant (phased value iteration) of $Q$-Learning is given as follows:

**Theorem A1.** *(Sample Complexity) Let $r \in [0, R_{\max}]$ be bounded reward, for an appropriate choice of $m$, the phased value iteration algorithm with peer reward $\tilde{r}_{\mathrm{peer}}$ calls the generative model $G(\widetilde{\mathcal{M}})$ $O\left(\frac{|\mathcal{S}||\mathcal{A}|T}{\epsilon^2(1-e_- -e_+)^2} \log \frac{|\mathcal{S}||\mathcal{A}|T}{\delta}\right)$ times in $T$ epochs, and returns a policy such that for all state $s \in \mathcal{S}$, $\left|\frac{1}{\eta}V^\pi(s) - V^*(s)\right| \leq \epsilon$, w.p. $\geq 1 - \delta$, $0 < \delta < 1$, where $\eta = 1 - e_- - e_+ > 0$ is a constant.*

*Proof.* Similar to Theorem 1, we firstly construct a transformed MDP $\hat{\mathcal{M}}$ and the optimal policies for these two MDP are equivalent (Lemma A1). As a result, we could analyse the sample complexity of phased value iteration under $\hat{\mathcal{M}}$.

It is easy to obtain that $\tilde{r}_{\mathrm{peer}} \in [0, R_{\max}]$ and $V^\pi(s) \in \left[0, \frac{R_{\max}}{1-\gamma}\right]$ are also bounded. Using Hoeffding's inequality, we have

$$\Pr\left(\left|\mathbb{E}\left[\hat{V}_{t+1}^*(s_{t+1})\right] - \sum_{s_{t+1}\in\mathcal{S}}\bar{\mathbb{P}}_a(s_t, s_{t+1})\hat{V}_{t+1}^*(s_{t+1})\right| \geq \epsilon\right) \leq 2\exp\left(\frac{-2m\epsilon^2(1-\gamma)^2}{R_{\max}^2}\right),$$

$$\Pr\left(\left|\mathbb{E}\left[\tilde{r}_{\mathrm{peer}}(s_t, a)\right] - \sum_{s_{t+1}\in\mathcal{S}}\hat{\mathbb{P}}_a(s_t, s_{t+1})\tilde{r}_{\mathrm{peer}}(s_t, a)\right| \geq \epsilon\right) \leq 2\exp\left(\frac{-2m\epsilon^2}{R_{\max}^2}\right).$$

Then the difference between learned value function $V^\pi(s)_t$ and optimal value function $\hat{V}^*(s)_t$ under transformed MDP at iteration $t$ is given:

$$\left|\hat{V}_t^*(s) - V_t(s)\right| = \max_{a\in\mathcal{A}}\mathbb{E}\left[r_t + \gamma V_{t+1}^*(s_{t+1})\right] - \max_{a\in\mathcal{A}}\sum_{s_{t+1}\in\mathcal{S}}\bar{\mathbb{P}}_a(s_t, s_{t+1})\left[\tilde{r}_{\mathrm{peer}}(s_t, a) + \gamma V_{t+1}(s_{t+1})\right]$$

$$\leq \max_{a\in\mathcal{A}}\left|\mathbb{E}\left[r_t\right] - \sum_{s_{t+1}\in\mathcal{S}}\bar{\mathbb{P}}_a(s_t, s_{t+1})\tilde{r}_{\mathrm{peer}}(s_t, a)\right|$$

$$+ \gamma\max_{a\in\mathcal{A}}\left|\mathbb{E}\left[\hat{V}_{t+1}^*(s_{t+1})\right] - \sum_{s_{t+1}\in\mathcal{S}}\bar{\mathbb{P}}_a(s_t, s_{t+1})V_{t+1}(s_{t+1})\right|$$

$$\leq \epsilon_1 + \max_{a\in\mathcal{A}}\left|\mathbb{E}\left[r_t\right] - \mathbb{E}\left[\tilde{r}_{\mathrm{peer}}\right]\right| + \gamma\epsilon_2 + \left|\mathbb{E}\left[\hat{V}_{t+1}^*(s_{t+1})\right] - \mathbb{E}\left[V_{t+1}(s_{t+1})\right]\right|$$

$$\leq \gamma\max_{s\in\mathcal{S}}\left|\hat{V}_{t+1}^*(s) - V_{t+1}(s)\right| + \epsilon_1 + \gamma\epsilon_2$$

Recursing above equation, we get

$$\max_{s\in\mathcal{S}}\left|\hat{V}^*(s) - V(s)\right| \leq (\epsilon_1 + \gamma\epsilon_2) + \gamma(\epsilon_1 + \gamma\epsilon_2) + \cdots + \gamma^{T-1}(\epsilon_1 + \gamma\epsilon_2)$$

$$= \frac{(\epsilon_1 + \gamma\epsilon_2)(1 - \gamma^T)}{1 - \gamma}$$

Let $\epsilon_1 = \epsilon_2 = \frac{(1-\gamma)\epsilon}{(1+\gamma)}$, then $\max_{s\in\mathcal{S}}\left|\hat{V}^*(s) - V(s)\right| \leq \epsilon$. In other words, for arbitrarily small $\epsilon$, by choosing $m$ appropriately, there always exists $\epsilon_1$ and $\epsilon_2$ such that the value function error is bounded within $\epsilon$. As a consequence the *phased value iteration* algorithm can converge to the near optimal policy within finite steps using peer reward.

Note that there are in total $|\mathcal{S}||\mathcal{A}|T$ transitions under which these conditions must hold, where $|\cdot|$ represent the number of elements in a specific set. Using a union bound, the probability of failure in any condition is smaller than

$$2|\mathcal{S}||\mathcal{A}|T \cdot \exp\left(-m\frac{\epsilon^2(1-\gamma)^2}{(1+\gamma)^2} \cdot \frac{(1-\gamma)^2}{R_{\max}^2}\right).$$

We set above failure probability less than $\delta$, and $m$ should satisfy that

$$m = O\left(\frac{1}{\epsilon^2}\log\frac{|\mathcal{S}||\mathcal{A}|T}{\delta}\right).$$

In consequence, after $m|\mathcal{S}||\mathcal{A}|T$ calls, which is, $O\left(\frac{|\mathcal{S}||\mathcal{A}|T}{\epsilon^2}\log\frac{|\mathcal{S}||\mathcal{A}|T}{\delta}\right)$, the value function converges to the optimal value function $\hat{V}^*(s)$ for every $s$ in transformed MDP $\widetilde{M}$, with probability greater than $1 - \delta$.

From Lemma A1, we know $\hat{V}^*(s) = (1 - e_- - e_+) \cdot V^*(s) + C$, where $C$ is a constant. Let $\epsilon = (1 - e_- - e_+) \cdot \epsilon'$ and $V(s) = (1 - e_- - e_+) \cdot V'(s) + C$, we have

$$|V^*(s) - V'(s)| = \left| \frac{\hat{V}^*(s) - C}{(1 - e_- - e_+)} - \frac{V(s) - C}{(1 - e_- - e_+)} \right| \tag{22}$$

$$= \frac{1}{(1 - e_- - e_+)}\left|\hat{V}^*(s) - V(s)\right| \le \epsilon' \tag{23}$$

This indicates that when the algorithm converges to the optimal value function for transformed MDP $\hat{\mathcal{M}}$, it also finds a underlying value function $V'(s) = \frac{1}{\eta}V(s)$ that converges the optimal value function $V^*(s)$ for original MDP $\mathcal{M}$.

As a consequence, we know it needs to call $\mathcal{O}\left(\frac{|\mathcal{S}||\mathcal{A}|T}{\epsilon'^2(1 - e_- - e_+)^2}\log\frac{|\mathcal{S}||\mathcal{A}|T}{\delta}\right)$ to achieve an $\epsilon'$ error in value function for original MDP $\mathcal{M}$, which is no more than $\mathcal{O}\left(\frac{1}{(1 - e_- - e_+)^2}\right)$ times of the one needed when the RL agent observes true rewards perfectly. When the noise is in high-regime, the algorithm suffers from a large $\frac{1}{(1 - e_- - e_+)^2}$ thus less efficient. Moreover, the sample complexity of phased value iteration with peer reward is equivalent to the one with surrogate reward in Wang et al. (2020) though sampling peer reward is less expensive and does not rely on any knowledge of noise rates. □

### A.3 MULTI-OUTCOME EXTENSION

In this section, we show our peer reward is generalizable to multi-class setting. Recall that in Section 3.2 we suppose the reward is discrete and has $|\mathcal{R}|$ levels, and the noise rates are characterized as $\mathbf{C}^{\mathrm{RL}}_{|\mathcal{R}|\times|\mathcal{R}|}$. Here we make further assumptions on the confusion matrix: the reward is misreported to each level with specific probability, e.g.,

$$\mathbf{C}^{\mathrm{RL}}_{|\mathcal{R}|\times|\mathcal{R}|} = \begin{bmatrix} 1 - \sum_{i\neq1}e_i, & e_2, & \cdots & e_{|\mathcal{R}|} \\ e_1, & 1 - \sum_{i\neq2}e_i, & \cdots & e_{|\mathcal{R}|} \\ \vdots & \cdots & \ddots & \vdots \\ e_1, & e_2, & \cdots, & 1 - \sum_{i\neq|\mathcal{R}|}e_i \end{bmatrix} \tag{24}$$

Following the notations in A.1, we define the peer reward in multi-outcome settings as $r(s, a) = \tilde{r}(s, a) - r'$, where $r'$ is randomly sampled following a specific sample policy $\pi_{\mathrm{sample}}$ over all state-action pairs. Let $\widetilde{R}_{\mathrm{peer}}, R, \widetilde{R}$, and $R'$ denote the random variables corresponding to $\tilde{r}_{\mathrm{peer}}, r, \tilde{r}$,

$r'$, $c_{ij}$ represents the entry of $\mathbf{C}^{\text{RL}}_{|\mathcal{R}| \times |\mathcal{R}|}$. Then we have

$$
\begin{aligned}
\mathbb{E}_{\pi}\left[\widetilde{R}\right] &= \sum_{i=1}^{|\mathcal{R}|} \mathbb{P}\left(R = R_i | \pi\right) \sum_{j=1}^{|\mathcal{R}|} c_{ij} R_j \\
&= \sum_{i=1}^{|\mathcal{R}|} \mathbb{P}\left(R = R_i | \pi\right) \left[ \left(1 - \sum_{j \neq i} e_i\right) R_i + \sum_{j \neq i} e_j R_j \right] \\
&= \sum_{i=1}^{|\mathcal{R}|} \mathbb{P}\left(R = R_i | \pi\right) \left[ \left(1 - \sum_{j=1}^{|\mathcal{R}|} e_i\right) R_i + \sum_{j=1}^{|\mathcal{R}|} e_j R_j \right] \\
&= \left(1 - \sum_{j=1}^{|\mathcal{R}|} e_j\right) \mathbb{E}_{\pi}\left[R\right] + \sum_{j=1}^{|\mathcal{R}|} e_j R_j,
\end{aligned}
$$

and

$$
\begin{aligned}
\mathbb{E}_{\pi_{\text{sample}}}\left[\widetilde{R}'\right] &= \sum_{i=1}^{|\mathcal{R}|} R_i \cdot \mathbb{P}\left(\widetilde{R} = R_i | \pi_{\text{sample}}\right) \\
&= \sum_{j=1}^{|\mathcal{R}|} R_j \sum_{i=1}^{|\mathcal{R}|} \mathbb{P}\left(R = R_i | \pi_{\text{sample}}\right) c_{ij} \\
&= \sum_{j=1}^{|\mathcal{R}|} R_j \left[ \sum_{i \neq j} \mathbb{P}\left(R = R_i | \pi_{\text{sample}}\right) e_j + \mathbb{P}\left(R = R_j | \pi_{\text{sample}}\right) \left(1 - \sum_{i \neq j} e_i\right) \right] \\
&= \sum_{j=1}^{|\mathcal{R}|} R_j \left[ \sum_{i=1}^{|\mathcal{R}|} \mathbb{P}\left(R = R_i | \pi_{\text{sample}}\right) e_j + \mathbb{P}\left(R = R_j | \pi_{\text{sample}}\right) \left(1 - \sum_{i=1}^{|\mathcal{R}|} e_i\right) \right] \\
&= \left(1 - \sum_{i=1}^{|\mathcal{R}|} e_i\right) \mathbb{E}_{\pi_{\text{sample}}}\left[R\right] + \sum_{j=1}^{|\mathcal{R}|} e_j R_j.
\end{aligned}
$$

Then, the peer reward is formulated as

$$
\begin{aligned}
\mathbb{E}\left[\widetilde{R}_{\text{peer}}\right] &= \mathbb{E}_{\pi}\left[\widetilde{R}\right] - \mathbb{E}\left[\widetilde{R}'\right] \\
&= \left(1 - \sum_{j=1}^{|\mathcal{R}|} e_j\right) \mathbb{E}_{\pi}\left[R\right] - \left(1 - \sum_{i=1}^{|\mathcal{R}|} e_i\right) \mathbb{E}_{\pi_{\text{sample}}}\left[R\right] \\
&= \left(1 - \sum_{j=1}^{|\mathcal{R}|} e_j\right) \mathbb{E}_{\pi}\left[R\right] + \text{const.}
\end{aligned}
$$

## A.4 EXTENSION IN MODERN DRL ALGORITHMS

In this section, we give the following deep reinforcement learning algorithms combined with our peer reward in Algorithm A3 and A4. In Algorithm A3, we give the peer reward aided robust policy gradient algorithm, where the gradient in Equation 25 corresponds to the loss function $\ell((s, a), q) = q \log \pi_\theta(a|s)$, which is classification calibrated (Liu & Guo, 2020). So the expectation of the gradient in 25 is an unbiased esitmation of the policy gradient in corresponding clean MDP. In (A4), we present a robust DQN algorithm with peer sampling, in which the origin loss is $\ell((s, a), \tilde{y})$, also classification calibrated. Thus the robustness can be proved via Liu & Guo (2020).

---

**Algorithm A3** Policy Gradient (Sutton et al., 1999) with Peer Reward

---

**Require:** $\widetilde{\mathcal{M}} = (\mathcal{S}, \mathcal{A}, \widetilde{\mathcal{R}}, \mathcal{P}, \gamma)$, learning rate $\alpha \in (0, 1)$, initial state distribution $\beta_0$, weight parameter $\xi$.
1: Initialize $\pi_\theta : \mathcal{S} \times \mathcal{A} \to \mathbb{R}$ arbitrarily
2: **for** $episode = 1$ **to** $M$ **do**
3:   Collect trajectory $\tau_\theta = \{(s_i, a_i, \tilde{r}_i)\}_{i=0}^T$, where $s_0 \sim \beta_0$, $a_t \sim \pi_\theta(\cdot|s_t)$, $s_{t+1} \sim \mathcal{P}(\cdot|s_t, a_t)$.
4:   Compute $q_t = \sum_{i=t}^T \gamma^{t-i} \tilde{r}_i$ for all $t \in \{0, 1, \dots, T\}$
5:   For each index $i \in \{0, 1, \dots, T\}$, we independently sample another two different indices $j, k$,
6:   and update policy parameter $\theta$ following

$$\theta \leftarrow \theta + \alpha \left[ q_i \nabla_\theta \log \pi_\theta(a_i|s_i) - \xi \cdot q_k \nabla_\theta \log \pi_\theta(a_j|s_j) \right] \tag{25}$$

7: **end for**
8: **return** $\pi_\theta$

---

**Algorithm A4** Deep $Q$-Network (Mnih et al., 2013) with Peer Reward

---

**Require:** $\widetilde{\mathcal{M}} = (\mathcal{S}, \mathcal{A}, \widetilde{\mathcal{R}}, \mathcal{P}, \gamma)$, learning rate $\alpha \in (0, 1)$, initial state distribution $\beta_0$, weight parameter $\xi$.
1: Initialize replay memory $\mathcal{D}$ to capacity $N$
2: Initialize action-value function $Q$ with random weights
3: **for** episode $= 1$ **to** $M$ **do**
4:   **for** $t = 1$ **to** $T$ **do**
5:    With probability $\epsilon$ select a random action $a_t$, otherwise select $a_t = \max_a Q^*(s, a)$
6:    Execute action $a_t$ and observe reward $\tilde{r}_t$ and observation $s_{t+1}$
7:    Store transition $(s_t, a_t, \tilde{r}_t, s_{t+1})$ in $\mathcal{D}$
8:    Sample three random minibatches of transitions $(s_i, a_i, \tilde{r}_i, s_{i+1})$, $(s_j, a_j, \tilde{r}_j, s_{j+1})$, $(s_k, a_k, \tilde{r}_k, s_{k+1})$ from $\mathcal{D}$.
9:    Set $\tilde{y}_i = \begin{cases} \tilde{r}_i & \text{for terminal} s_i \\ \tilde{r}_i + \gamma \max_{a'} Q(s_{i+1}, a') & \text{for non-terminal } s_{i+1} \end{cases}$
10:    Set $\tilde{y}_{\text{peer}} = \begin{cases} \tilde{r}_k & \text{for terminal} s_i \\ \tilde{r}_k + \gamma \max_{a'} Q(s_{j+1}, a') & \text{for non-terminal } s_{j+1} \end{cases}$
11:    Perform a gradient descent step on $(\tilde{y}_i - Q(s_i, a_i))^2 - \xi \cdot (\tilde{y}_{\text{peer}} - Q(s_j, a_j))^2$
12:   **end for**
13: **end for**
14: **return** $Q$

---

## B  ANALYSIS OF PEERBC

We prove that the policy learned by PeerBC converges to the expert policy when observing a sufficient amount of weak demonstrations in Theorem A2.

**Theorem A2.** *With probability at least $1 - \delta$, the error rate is upper-bounded by*

$$R^*_{D_E} \leq \frac{1 + \xi}{1 - e_- - e_+} \sqrt{\frac{2 \log 2/\delta}{N}}, \tag{26}$$

*where $N$ is the number of state-action pairs demonstrated by the expert.*

*Proof.* Recall $\widetilde{\mathcal{D}}_E$ denotes the joint distribution of imperfect expert' state-action pair $(s, \tilde{a})$. Assume there is a perfect expert and the corresponding state-action pairs $(s, a) \sim \mathcal{D}_E$. The indicator classification loss $\mathbb{1}(\pi(s), a)$ is specified here for a clean presentation, where $\mathbb{1}(\pi(s), a) = 1$ when $\pi(s) \neq a$, otherwise $\mathbb{1}(\pi(s), a) = 0$. Let $\widetilde{D}_E := \{(s_i, \tilde{a}_i)\}_{i=1}^N$ be the set of imperfect demonstrations, and $D_E := \{(s_i, \tilde{a}_i)\}_{i=1}^N$ be the set of weak demonstrations. Define:

$$R_{\mathcal{D}_E}(\pi) := \mathbb{E}_{(s,a) \sim \mathcal{D}_E} [\mathbb{1}(\pi(s), a)], \ R_{\widetilde{\mathcal{D}}_E}(\pi) := \mathbb{E}_{(s,\tilde{a}) \sim \mathcal{D}_E} [\mathbb{1}(\pi(s), \tilde{a})]$$

$$\hat{R}_{D_E}(\pi) := \frac{1}{N} \sum_{i \in [N]} \mathbb{1}(\pi(s_i), a_i), \ \hat{R}_{\widetilde{D}_E}(\pi) := \frac{1}{N} \sum_{i \in [N]} \mathbb{1}(\pi(s_i), \tilde{a}_i).$$

Note we focus on the analyses of loss in this proof. The negative of loss can be seen as a reward. Denote by $\pi_{\widetilde{D}_E}$ and $\pi_{\widetilde{\mathcal{D}}_E}$ be the optimal policy obtained with minimizing the indicator loss with dataset $\widetilde{D}_E$ and distribution $\widetilde{\mathcal{D}}_E$. We shorten $\pi_{\widetilde{D}_E}$ as $\tilde{\pi}^*$, which is the best policy we can learn from imperfect demonstration with our algorithm. Let $\pi^*$ be the policy for the perfect expert. We would like to see the performance gap of policy learning between imperfect demonstrations and perfect demonstrations, i.e. $R_{\mathcal{D}_E}(\tilde{\pi}^*) - R_{\mathcal{D}_E}(\pi^*)$. Using Hoeffding's inequality with probability at least $1 - \delta$, we have

$$|\hat{R}_{\widetilde{D}_E}(\pi) - R_{\widetilde{\mathcal{D}}_E}(\pi)| \leq (1 + \xi)\sqrt{\frac{\log 2/\delta}{2N}}.$$

Note we also have

$$R_{\widetilde{\mathcal{D}}_E}(\tilde{\pi}^*) - R_{\widetilde{\mathcal{D}}_E}(\pi_{\widetilde{\mathcal{D}}_E})$$

$$\leq \hat{R}_{\widetilde{D}_E}(\tilde{\pi}^*) - \hat{R}_{\widetilde{D}_E}(\pi_{\widetilde{\mathcal{D}}_E}) + \left( R_{\widetilde{\mathcal{D}}_E}(\tilde{\pi}^*) - \hat{R}_{\widetilde{D}_E}(\tilde{\pi}^*) \right)$$

$$+ \left( \hat{R}_{\widetilde{D}_E}(\pi_{\widetilde{\mathcal{D}}_E}) - R_{\widetilde{\mathcal{D}}_E}(\pi_{\widetilde{\mathcal{D}}_E}) \right)$$

$$\leq 0 + 2 \max_{\pi} \left| \hat{R}_{\widetilde{D}_E}(\pi) - R_{\widetilde{\mathcal{D}}_E}(\pi) \right|$$

$$\leq (1 + \xi)\sqrt{\frac{2 \log 2/\delta}{N}}.$$

Before proceeding, we need to define a constant to show the affect of label noise. When the dimension of action space is 2, the problem is essentially a binary classification with noisy labels (Liu & Guo, 2020), where the noise rate (a.k.a confusion matrix) is defined as $e_+ = \mathbb{P}(\tilde{\pi}_E(s) = A_- | \pi^*(s) = A_+)$ and $e_- = \mathbb{P}(\tilde{\pi}_E(s) = A_+ | \pi^*(s) = A_-)$. Recall the action space is defined as $\mathcal{A} = \{A_+, A_-\}$. The noise constant is denoted by $e = e_{-1} + e_{+1}$. Accordingly, when the dimension of action space is $|\mathcal{R}| > 2$, we can also get similar results under uniform noise where

$$e_u := \mathbb{P}(\tilde{\pi}_E(s) = u | \pi^*(s) = u'), u' \neq u. \tag{27}$$

The noise constant $e$ is denoted by $e = \sum_{u=1}^{|\mathcal{R}|} e_u$. The feature-independent assumption holds thus the properties of peer loss functions (Liu & Guo, 2020) can be used, i.e.

$$R_{\mathcal{D}_E}(\tilde{\pi}^*) - R_{\mathcal{D}_E}(\pi^*)$$

$$= \frac{1}{1 - e} \left( R_{\widetilde{\mathcal{D}}_E}(\tilde{\pi}^*) - R_{\widetilde{\mathcal{D}}_E}(\pi_{\widetilde{\mathcal{D}}_E}) \right)$$

$$\leq \frac{1 + \xi}{1 - e} \sqrt{\frac{2 \log 2/\delta}{N}}$$

From definition and deterministic assumption for $\pi^*$, we have $R_{\mathcal{D}_E}(\pi^*) = 0$. Thus the error rate in the $k$-th iteration is

$$
\begin{aligned}
R_{\mathcal{D}_E}(\tilde{\pi}^*) &\leq R_{\mathcal{D}_E}(\pi^*) + \frac{1+\xi}{1-e}\sqrt{\frac{2\log 2/\delta}{N}} \\
&= \frac{1+\xi}{1-e}\sqrt{\frac{2\log 2/\delta}{N}}.
\end{aligned}
\tag{28}
$$

Note $R_{\mathcal{D}_E}(\tilde{\pi}^*) = R_{\widetilde{\mathcal{D}}_E}$ by definition. $\qquad\square$

## C    SUPPLEMENTARY EXPERIMENTS

### C.1    EXPERIMENTAL SETUP

We set up our experiments within the popular OpenAI `stable-baselines`[2] and `keras-rl`[3] framework. Specifically, three popular RL algorithms including Deep-$Q$-Network (DQN) (Mnih et al., 2013; van Hasselt et al., 2016), Dueling-DQN (DDQN) (Wang et al., 2016) and Proximal Policy Optimization Algorithms (PPO) are evaluated in a varied of OpenAI Gym environments including classic control games (`CartPole`, `Acrobot`) and vision-based Atari-2600 games (`Breakout`, `Boxing`, `Enduro`, `Freeway`, `Pong`).

### C.2    IMPLEMENTATION DETAILS

**RL with noisy reward**    Following Wang et al. (2020), we consider the binary reward $\{-1, 1\}$ for Cartpole where the symmetric noise is synthesized with different error rates $e = e_- = e_+$. We adopted a five-layer fully connected network and the Adam optimizer. The model is trained for 10,000 steps with the learning rate of $1e^{-3}$ and the Boltzmann exploration strategy. The update rate of target model and the memory size are $1e^{-2}$ and 50,000. The performance is reported under 10 independent trials with different random seeds.

**BC with weak expert**    We train the imperfect expert on the framework `stable-baselines` with default network architecture for Atari and hyper-parameters from `rl-baselines-zoo`[4]. The expert model is trained for $1,400,000$ steps for Pong and $2,000,000$ steps for Boxing, Enduro and Freeway. For each of those environment, We use the trained model to generate 100 trajectories, and behavior cloning is performed on these trajectories. We adopt cross entropy loss for behavior cloning and add a small constant ($1 \times 10^{-8}$) for each logit after the softmax operation for peer term to avoid this term become too large. In BC experiments, the batchsize is 128, learning rate is $1 \times 10^{-4}$ and the $\epsilon$ value for Adam optimizer is $1 \times 10^{-8}$.

**Policy co-training**    For the experiments on Gym (CartPole and Acrobot), we mask the first co-ordinate in the state vector for one view and the second for the other, same as Song et al. (2019). Both policies are trained with PPO(Schulman et al., 2017) + PeerBC. In each iteration, we sample 128 steps from each of the 8 parallel environments. These samples are fed to PPO training with a batchsize of 256, a learning rate of $2.5 \times 10^{-4}$ and a clip range of 0.1. Both learning rate and clip range decay to 0 throughout time. We represent the policy by a fully connected network with 2 hidden layers, each has 128 units.

For the experiments on Atari (Pong and Breakout), the input is raw game images. We adopt the preprocess introduced in Mnih et al. (2013) and mask the pixels in odd columns for one view and even columns for the other. The policy we use adopts a default CNN as in `stable-baselines`. Batchsize, learning rate, clip range and other hyper-parameters are the same as Gym experiments. Note that we only add PeerBC after 1000 episodes.

---

[2]`https://github.com/hill-a/stable-baselines`
[3]`https://github.com/keras-rl/keras-rl`
[4]`https://github.com/araffin/rl-baselines-zoo/blob/master/hyperparams/ppo2.yml#L1`

## C.3 SUPPLEMENTARY RESULTS FOR FIGURE 2 AND TABLE 1

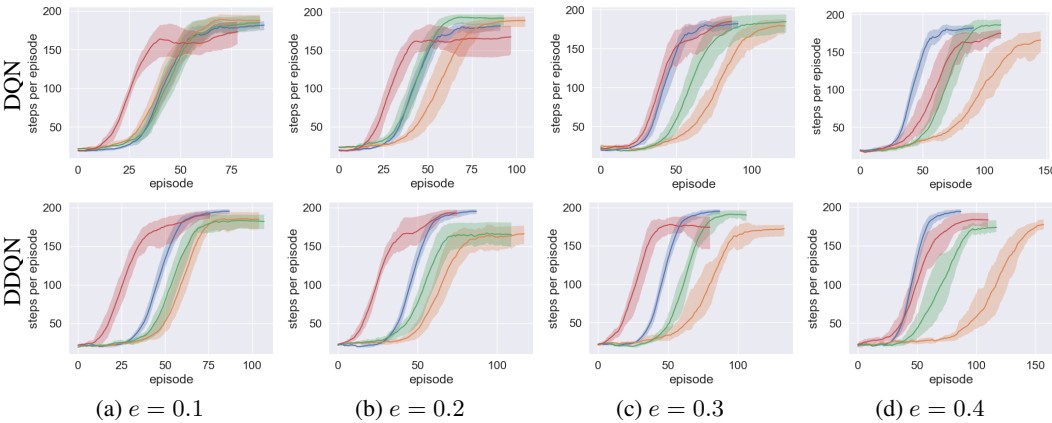

Figure A1: Learning curves on CartPole game with true reward ($r$) ■, noisy reward ($\tilde{r}$) ■, surrogate reward (Wang et al., 2020) ($\hat{r}$) ■, and peer reward ($\tilde{r}_{\text{peer}}$, $\xi = 0.2$) ■. Each experiment is repeated 10 times with different random seeds.

Table A1: Numerical performance of DDQN on CartPole with true reward ($r$), noisy reward ($\tilde{r}$), surrogate reward $\hat{r}$ (Wang et al., 2020), and peer reward $\tilde{r}_{\text{peer}}(\xi = 0.2)$. $\mathcal{R}_{avg}$ denotes average reward per episode after convergence, (last five episodes) the higher ($\uparrow$) the better; $N_{epi}$ denotes total episodes involved in 10,000 steps, the lower ($\downarrow$) the better.

| | | $e = 0.1$ | | $e = 0.2$ | | $e = 0.3$ | | $e = 0.4$ | |
|---|---|---|---|---|---|---|---|---|---|
| | | $\mathcal{R}_{avg} \uparrow$ | $N_{epi} \downarrow$ | $\mathcal{R}_{avg} \uparrow$ | $N_{epi} \downarrow$ | $\mathcal{R}_{avg} \uparrow$ | $N_{epi} \downarrow$ | $\mathcal{R}_{avg} \uparrow$ | $N_{epi} \downarrow$ |
| DQN | $r$ | $183.6 \pm 7.6$ | $101.3 \pm 4.8$ | $184.0 \pm 7.3$ | $101.5 \pm 4.6$ | $184.0 \pm 7.3$ | $101.5 \pm 4.6$ | $184.0 \pm 7.3$ | $101.5 \pm 4.6$ |
| | $\tilde{r}$ | $\mathbf{189.3 \pm 12.7}$ | $98.2 \pm 6.5$ | $189.7 \pm 7.9$ | $110.5 \pm 7.1$ | $183.2 \pm 9.8$ | $130.5 \pm 7.7$ | $169.7 \pm 18.6$ | $150.2 \pm 11.4$ |
| | $\hat{r}$ | $188.3 \pm 8.2$ | $101.1 \pm 6.2$ | $\mathbf{192.7 \pm 9.2}$ | $97.9 \pm 6.4$ | $185.4 \pm 15.9$ | $116.9 \pm 11.0$ | $\mathbf{184.8 \pm 16.4}$ | $123.1 \pm 8.6$ |
| | $\tilde{r}_{\text{peer}}$ | $177.2 \pm 19.1$ | $\mathbf{91.2 \pm 5.9}$ | $170.0 \pm 24.8$ | $\mathbf{94.6 \pm 8.5}$ | $\mathbf{190.5 \pm 14.3}$ | $\mathbf{99.4 \pm 5.2}$ | $183.1 \pm 13.3$ | $\mathbf{118.1 \pm 10.7}$ |
| DDQN | $r$ | $195.6 \pm 3.1$ | $101.2 \pm 3.2$ | $195.6 \pm 3.1$ | $101.2 \pm 3.2$ | $195.6 \pm 3.1$ | $101.2 \pm 3.2$ | $195.2 \pm 3.0$ | $101.2 \pm 3.3$ |
| | $\tilde{r}$ | $185.2 \pm 15.6$ | $114.6 \pm 6.0$ | $168.8 \pm 13.6$ | $123.9 \pm 9.6$ | $177.1 \pm 11.2$ | $133.2 \pm 9.1$ | $185.5 \pm 10.9$ | $163.1 \pm 11.0$ |
| | $\hat{r}$ | $183.9 \pm 10.4$ | $110.6 \pm 6.7$ | $165.1 \pm 18.2$ | $113.9 \pm 9.6$ | $\mathbf{192.2 \pm 10.9}$ | $115.5 \pm 4.3$ | $179.2 \pm 6.6$ | $125.8 \pm 9.6$ |
| | $\tilde{r}_{\text{peer}}$ | $\mathbf{198.5 \pm 2.3}$ | $\mathbf{86.2 \pm 5.0}$ | $\mathbf{195.5 \pm 9.1}$ | $\mathbf{85.3 \pm 5.4}$ | $174.1 \pm 32.5$ | $\mathbf{88.8 \pm 6.3}$ | $\mathbf{191.8 \pm 8.5}$ | $\mathbf{106.9 \pm 9.2}$ |

## C.4 ADDITIONAL RESULTS FOR PEERRL ON PENDULUM

To further evaluate the effectiveness of PeerRL, we conduct experiments on a more challenging continuous control task *Pendulum*, where the goal is to keep a frictionless pendulum standing up. Since the rewards in pendulum are continuous: $r \in (-16.3, 0.0]$, we discretized it into 17 intervals: $(-17, -16], (-16, -15], \cdots, (-1, 0]$, with its value approximated using its maximum point. We experiment the DDPG (Lillicrap et al., 2015) and uniform noise in this environment. In Figure A3, the RL agents with proposed CA objective successfully converge to the optimal policy under different amounts of noise. On the contrary, the agents with noisy rewards suffer from the biased noise especially in a high-noise regime.

## C.5 SENSITIVITY ANALYSIS OF PEER PENALTY $\xi$

In this section, we analyze the sensitivity of $\xi$ in RL and BC tasks. Note that we did not tune this hyperparameter extensively in all the experiments presented above since we found our method works robustly in a wide range of $\xi$.

**RL with noisy reward** We repeat the experiment in Figure A1 for DQN but with a varying $\xi$ from 0.1 to 0.4. As shown in Figure A2, our method works reasonably and leads to faster convergence compared to baselines. However, we found that the late stage of training, a small $\xi$ is necessary since the agent already gains useful knowledge and make reasonable actions, therefore, an over-large

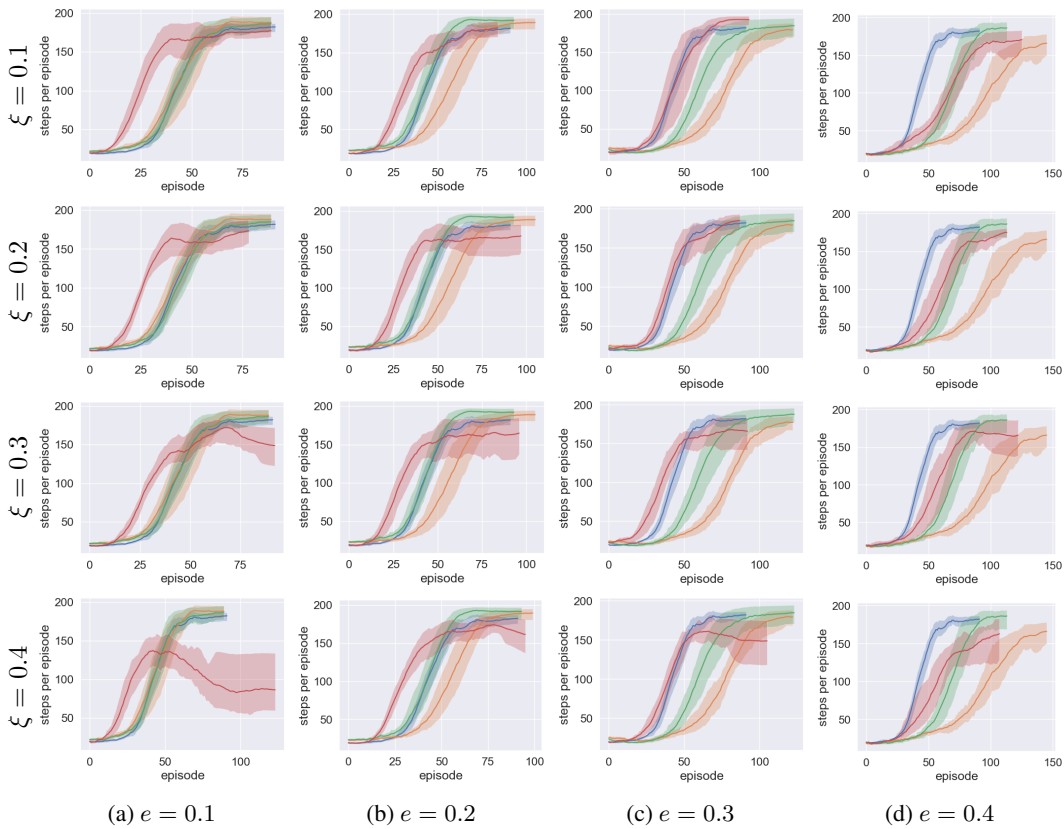

Figure A2: Learning curves of DQN on CartPole game with peer reward ($\tilde{r}_{\text{peer}}$) ■ under different choices of $\xi$ (from 0.1 to 0.4).

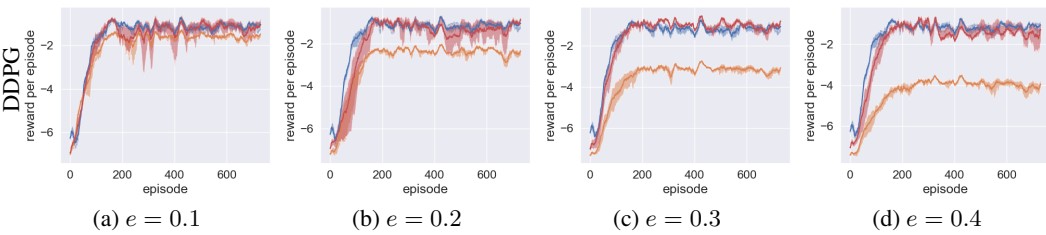

Figure A3: Learning curves of DDPG (Lillicrap et al., 2015) on Pendulum with true reward ($r$) ■, noisy reward ($\tilde{r}$) ■, and peer reward ($\tilde{r}_{\text{peer}}$) ■.

penalty might avoid the agent achieving simple agreements with the supervision signals, especially in a low-noise regime (see $\xi = 0.4, e = 0.1$). This observation inspires us that a decay schedule of $\xi$ might be helpful in stabilizing the training of PeerRL algorithms. To verify this hypothesis, we repeat the above experiments but with a linear decay $\xi$ that decreases from 0.4 to 0.1. In Figure A4, we found the linear decay schedule is able to stabilize the convergence of PeerRL algorithms compared to static $\xi = 0.4$. The theoretical principles and insights of dynamic peer penalty merit further study.

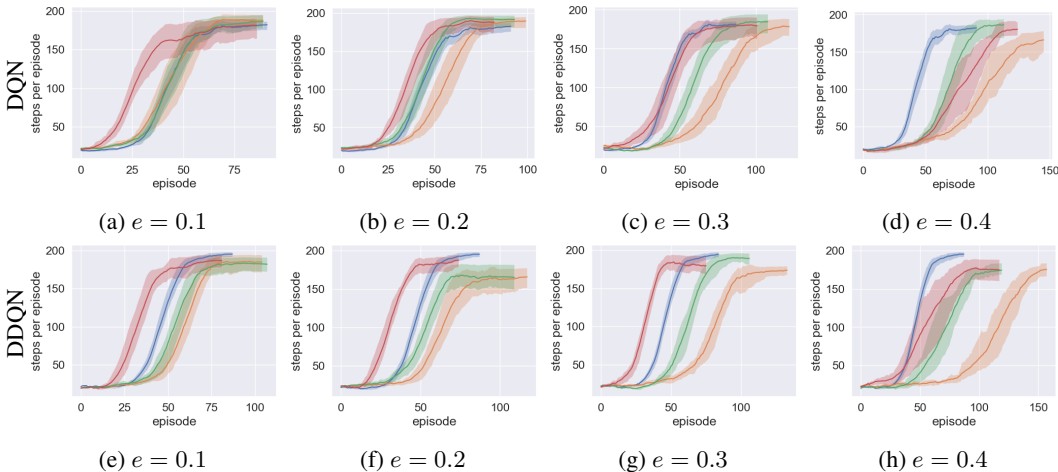

Figure A4: Learning curves of DQN on CartPole game with peer rewards ($\tilde{r}_{\text{peer}}$) ■. Here, a linear decay $\xi$ is applied during training procedure (initial $\xi = 0.4$). Compared to static $\xi = 0.4$, the linear decay peer penalty stabilizes the convergence of RL algorithms.

**BC from weak demonstrations**  We conduct experiments on Pong with 12 different $\xi$ values, varying from 0.1 to 1.2. As a reference, two other BC baselines GAIL (Ho & Ermon, 2016) and SQIL (Reddy et al., 2019) are considered. We do not include GAIL in the figure, since GAIL fails to produce meaningful results on vision-based Atari games as also observed (Reddy et al., 2019; Brantley et al., 2020). From Figure A5, we can see PeerBC outperforms pure behavior cloning and SQIL(Reddy et al., 2019) when $\xi$ is within $[0.1, 0.7]$, revealing our proposed PeerBC is a superior behavior cloning approach able to better elicit information from imperfect demonstrations.

## C.6  VARIANCE REDUCTION IN NOISY REWARD SETTING

In practice, there is a trade-off question between bias and variance that influence the RL training. Even though variance reduction approaches theoretically do not resolve the challenge when bias presents in the observed rewards, it might be beneficial to the training procedure in practice. To investigate whether the variance reduction techniques are helpful in the setting with biased noise, we repeated the experiments in Table 1. We adopt the variance reduction technique (VRT) proposed in Romoff et al. (2018) and found it brings little benefits to noisy reward and surrogate reward as shown in Figure A6. However, peer + VRT performs worse compared to peer reward only. This demonstrates the benefits of scenarios does not mainly come from the variance reduction in our setting. The full quantitative results are presented in Table A2.

## C.7  STOCHASTIC POLICY FOR BEHAVIORAL CLONING

In this section, we analyze the stochasticity of the imperfect expert model and fully-converged PPO agent (assumed to be the clean expert), and show that our PeerBC can handle both cases when the clean expert is stochastic and when it's rather deterministic.

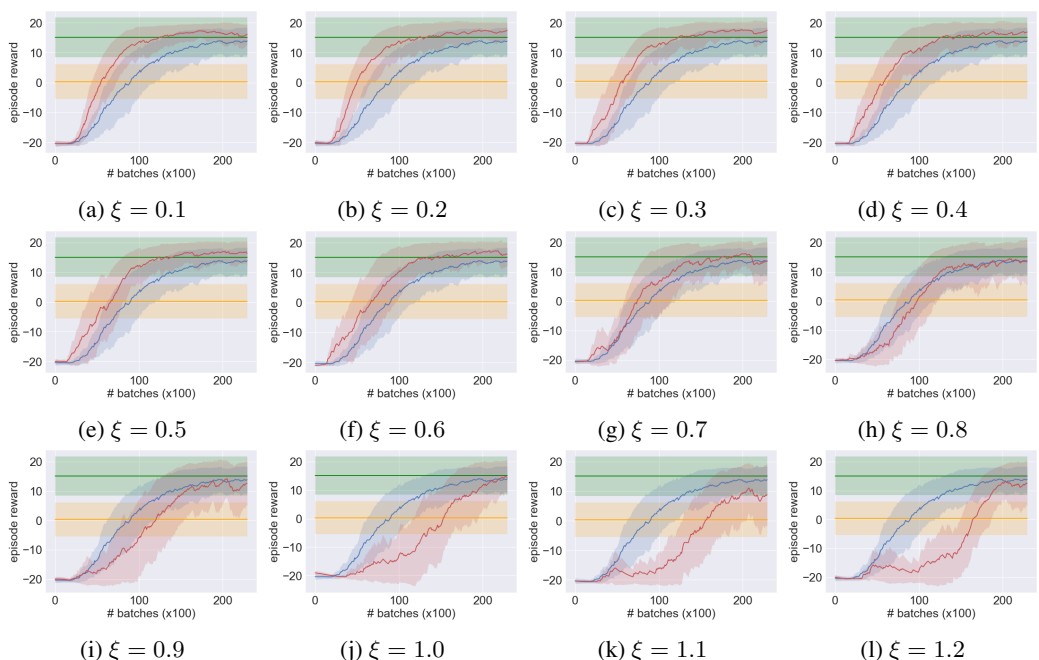

Figure A5: Sensitivity analysis of $\xi$ for PeerBC on Pong with behavior cloning ▉, PeerBC ▉ ($\xi$ varies from 0.2 to 0.5 and 1.0), expert ▉, and SQIL ▉ reported by SQIL (Reddy et al., 2019). Each experiment is repeated under 3 different random seeds.

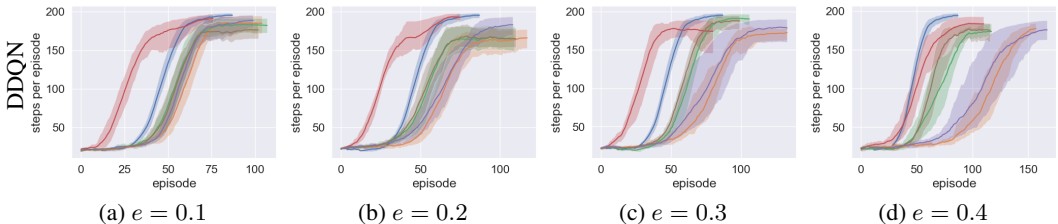

Figure A6: Learning curves of DDQN on CartPole with true reward ($r$) ▉, noisy reward ($\tilde{r}$) ▉, noisy reward + VRT ▉, surrogate reward (Wang et al., 2020) ($\hat{r}$) ▉, surrogate reward + VAT ▉ and peer reward ($\tilde{r}_{\text{peer}}$, $\xi = 0.2$) ▉.

Table A2: Numerical performance of DDQN on CartPole with true reward ($r$), noisy reward ($\tilde{r}$), surrogate reward $\hat{r}$ (Wang et al., 2020), and peer reward $\tilde{r}_{\text{peer}}(\xi = 0.2)$. + VRT denotes the variance reduction techniques in Romoff et al. (2018) are adopted.

| | | $e = 0.1$ | | $e = 0.2$ | | $e = 0.3$ | | $e = 0.4$ | |
|---|---|---|---|---|---|---|---|---|---|
| | | $\mathcal{R}_{avg} \uparrow$ | $N_{epi} \downarrow$ | $\mathcal{R}_{avg} \uparrow$ | $N_{epi} \downarrow$ | $\mathcal{R}_{avg} \uparrow$ | $N_{epi} \downarrow$ | $\mathcal{R}_{avg} \uparrow$ | $N_{epi} \downarrow$ |
| DDQN | $r$ | $195.6 \pm 3.1$ | $101.2 \pm 3.2$ | $195.6 \pm 3.1$ | $101.2 \pm 3.2$ | $195.6 \pm 3.1$ | $101.2 \pm 3.2$ | $195.2 \pm 3.0$ | $101.2 \pm 3.3$ |
| | $\tilde{r}$ | $185.2 \pm 15.6$ | $114.6 \pm 6.0$ | $168.8 \pm 13.6$ | $123.9 \pm 9.6$ | $177.1 \pm 11.2$ | $133.2 \pm 9.1$ | $185.5 \pm 10.9$ | $163.1 \pm 11.0$ |
| | +VRT | $190.2 \pm 14.6$ | $111.8 \pm 4.5$ | $183.6 \pm 15.7$ | $118.0 \pm 9.5$ | $179.6 \pm 19.9$ | $129.6 \pm 13.9$ | $182.6 \pm 13.4$ | $157.2 \pm 19.0$ |
| | $\hat{r}$ | $183.9 \pm 10.4$ | $110.6 \pm 6.7$ | $165.1 \pm 18.2$ | $113.9 \pm 9.6$ | $192.2 \pm 10.9$ | $115.5 \pm 4.3$ | $179.2 \pm 6.6$ | $125.8 \pm 9.6$ |
| | + VRT | $179.0 \pm 13.5$ | $112.1 \pm 5.4$ | $163.4 \pm 17.0$ | $112.8 \pm 9.1$ | $190.1 \pm 10.3$ | $112.6 \pm 3.9$ | $179.8 \pm 15.1$ | $119.7 \pm 9.1$ |
| | $\tilde{r}_{\text{peer}}$ | $198.5 \pm 2.3$ | $86.2 \pm 5.0$ | $195.5 \pm 9.1$ | $85.3 \pm 5.4$ | $174.1 \pm 32.5$ | $88.8 \pm 6.3$ | $191.8 \pm 8.5$ | $106.9 \pm 9.2$ |
| | + VRT | $184.1 \pm 8.8$ | $95.2 \pm 8.8$ | $193.9 \pm 6.0$ | $86.5 \pm 3.9$ | $174.2 \pm 24.3$ | $93.5 \pm 6.2$ | $181.2 \pm 15.3$ | $117.7 \pm 9.5$ |

We plot the entropy of the PPO agent during training on four environments from the BC task in Figure A7, and we give the entropy value of the imperfect expert model and the optimal policy in Table A3. We observe that except for Freeway, the entropy of expert policies is always larger than 1. We calculate the mean value of the highest action probability over 1000 steps for the full-converged PPO agents in Table A4, which again verifies that the true expert policy we aim to recover might not

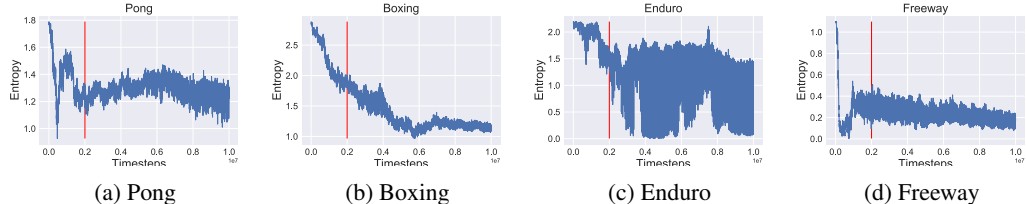

(a) Pong  (b) Boxing  (c) Enduro  (d) Freeway

Figure A7: The policy entropy of the PPO agent during training. The imperfect expert model is trained for $0.2 \times 10^7$ timesteps as the red line indicates.

be fully deterministic. These results demonstrate the flexibility of our proposed approach in dealing with both stochastic and deterministic clean expert policies in practice, although a deterministic clean expert policy is assumed in our theoretical analysis.

Also, from Figure A7 and Table A3, we notice that the entropy of imperfect expert models are higher than the fully converged PPO agents, implying that the expert models might contain an amount of noise. That's because there might be states on which the expert has not seen enough and the selected actions contain much noise. This is consistent with our claim, that the benefits of PeerBC might come from two aspects, both noise reduction of the imperfect expert and inducing a more deterministic policy.

| Timesteps ($\times 10^7$) | Pong | Boxing | Enduro | Freeway |
|---|---|---|---|---|
| 0.2 (Imperfect Expert) | 1.201 | 1.949 | 1.637 | 0.318 |
| 1.0 (Fully converged PPO) | 1.250 | 1.168 | 1.126 | 0.171 |

Table A3: The policy entropy of the PPO agent during training.

| Trained timesteps ($\times 10^7$) | Pong | Boxing | Enduro | Freeway |
|---|---|---|---|---|
| 1.0 (Fully converged PPO) | 0.492 | 0.579 | 0.664 | 0.903 |

Table A4: The mean value of the highest action probability over 1000 steps.

