# OpenReview forum: "Policy Learning Using Weak Supervision"
_ICLR.cc/2021/Conference — Reject_

### Official Review · AnonReviewer4 · 2020-10-23
**Well written and presents simple idea, but with some concerns**

**Rating:** 6
**Confidence:** 3

**Review:**

This paper tackles the problem of policy learning under weak/noisy supervision. The authors present PeerPL, a unified framework that can train agents using behavior cloning under noisy/suboptimal demonstrations, or using reinforcement learning under noisy rewards. PeerPL uses the idea of “Correlated Agreement” by subtracting the original objective with a second term. The second term evaluates on randomly paired state-action tuples and supervisions, punishing the “blind” agreement between the learning agent and the weak supervision. The authors instantiate this idea on both behavior cloning and RL (PG/DQN), and also evaluate it on the problem of policy co-training. The authors demonstrate that PeerPL outperforms the weak supervision baselines on IL and RL setting, and sometimes it even outperforms agents trained with clean supervision.


Strengths

+ The paper overall is well-written and easy to follow
+ The proposed idea is simple and general, and can be applied to both RL and IL
+ The proposed method does not require prior knowledge on the noise structure
+ Nice and sound convergence analysis
+ Proposed method outperform baselines in most tasks under both BC/RL settings and policy-cotraining


Weaknesses

- Although the experiments prove otherwise, the second term in the objective is not intuitive to me. The learning agent is not supposed to agree with randomly sampled supervision anyways, unless the noise is constructed in a way that encourages the agent to do so. I cannot convince myself that the second term is helpful other than potentially some normalizing factor that helps reduce the variance in the RL setting, or maybe help remove some misleading prior in the BC setting.

- I am confused by the results in Figure 2 that PeerPL outperforms agents trained with *clean* reward. I am not convinced by the authors’ explanation in Section 5.1, as subtracting the reward with randomly sampled {-1,1} will not help exploration.

- The theoretical results are not particularly useful as they assume binary reward/actions, which is rarely the case in practical settings.


Other Comments

- It would have been nice in section 5.2 how PeerBC compares against IRL, which is supposed to handle imperfect demonstrations to some degree.

- For PeerRL, it would be nice to see results on tasks that are harder than CartPole, for example, Atari or other continuous control tasks.

---

> ### Author Response · Authors · 2020-11-22
> **Response to AnonReviewer4**
>
> Q1: Objective is not intuitive, hard to understand, more explanation
>
> Our objective does not encourage the agent to agree with randomly sampled supervision. On the contrary, consider the objective with $\xi = 1$. The sign of the second term is the opposite of the first term. Thus, this objective discourages the agent to agree with random supervisions. Our theorem shows that the proposed objective is invariant to noise (Theorem 1) and could recover the clean policy (Theorem 1, 2). For the hypothesis of variance reduction, we conduct additional experiments and found little of the benefits come from the variance reduction. Please refer to Q2/A2 for R1.
>
> The intuition of the CA objective comes from information verification methods in the crowdsourcing literature. Assuming we have collected reports on multiple tasks from several reporters via crowdsourcing, correlated agreement (CA) is a mechanism that scores the reports from any two reporters $A$ and $B$ on the same task $i$ without accessing the ground truth. Let $S$ be a simple score function based on whether two reports agree (like $S(r_1, r_2) = 1$ if $r_1 = r_2$, and 0 otherwise). CA will first score the reports from the two reporters on the task $i$ with the score function $S$, i.e. $S(r_A^i, r_B^i)$, where superscript denotes the task and subscript denotes the reporter. Then CA samples two peer tasks $j, k$ and scores the reports from $A, B$ on them respectively, i.e. $S(r_A^j, r_B^k)$. The final score is $S(r_A^i, r_B^i) - S(r_A^j, r_B^k)$. This mechanism punishes blind agreement in which case the two reporters yield the same reports across all the tasks, which is non-informative but will receive a high score under only the first term $S(r_A^i, r_B^i)$. It has been proved that under mild assumptions, with the CA scoring function, the truth-telling strategy obtains the highest score.
>
> Back to our PeerPL, we treat our policy as one reporter, the weak supervision as another one, and our Eva function as the score function to get our CA objective. The truth-telling strategy corresponds to the optimal policy, which the CA objective will intuitively induce.
>
> We will add more explanations in the Appendix to make it easier to understand our objective.
>
> -----
>
> Q2: PeerPL outperforms the agents with true reward in Figure 2.
>
> Some works that reveal the connection between noise and exploration point out adding moderate noise into the environment or the neural network may benefit the exploration (Wang et al., 2020; Plappert et al., 2018; Fortunato et al., 2017). But too much noise will lead the training to fail. In our cases, the peer sampling technique cancels part of the noise to maintain it in a proper level, while the PeerPL can take advantage of the noisy reward to better explore. Due to the exploration, PeerPL can sometimes converge faster and yield better results than training with clean rewards.
>
> ------
>
> Q3: The theoretical results assume binary reward/actions, which is rarely the case in practical settings.
>
> Our theory analysis is not limited to the binary case (see Remark in Sec 4.2). We provided the theoretical results for the multi-outcome extension in Appendix A.3. Moreover, although we make assumptions on the noise model, our empirical results in Sec 5.2 show that our solution has the potential to deal with the noise in the real world (e.g., imperfect expert demonstrations in behavior cloning and peer agent’s demonstrations in co-training).
>
> -----
>
> Q4: It would have been nice in section 5.2 how PeerBC compares against IRL, which is supposed to handle imperfect demonstrations to some degree.
> As we discussed in appendix C.4, we tried IRL methods like GAIL, but GAIL fails to produce meaningful results on vision-based Atari games as also observed in (Reddy et al., 2019; Brantley et al., 2020). One possible reason is that IRL learns the reward function and the policy at the same time, which is harder to train compared with BC, especially on vision-based tasks, where learning features is more challenging. In contrast, our PeerBC is easy to implement and train, and needs no prior knowledge.
>
> -----
>
> Q5: For PeerRL, it would be nice to see results on tasks that are harder than CartPole, for example, Atari or other continuous control tasks.
>
> Thanks for the suggestion. We are working on the experiment on Pendulum (one continuous control task). We will post the results here when they are ready.
>
> -----
>
> [1] Reinforcement learning with perturbed reward. Wang et al., 2020.
>
> [2] Multi-goal reinforcement learning: Challenging robotics environments and request for research. Plappert et al., 2018.
>
> [3] Noisy networks for exploration. Fortunato et al., 2017.
>
> [4]  SQIL: Behavior Cloning via ReinforcementLearning with Sparse Rewards. Reddy et al., 2019.
>
> [5] Disagreement-Regularized Behavior Cloning. Brantley et al., 2020.

---

> > ### Comment · AnonReviewer4 · 2020-11-24
> > **Response to rebuttals**
> >
> > Thanks for the clarification, I have updated my score accordingly, and here the responses:
> >
> >
> > Q1.
> > I understand that the PeerPL objective does not encourage blind agreement, and how it punishes blind agreement. I would like to clarify what I meant by my original comments: "The learning agent is not supposed to agree with randomly sampled supervision anyways, unless the noise is constructed in a way to do so". What I meant is that an agent is not supposed to "blindly agree with randomly sampled labels", even trained with the “vanilla” objectives, unless the “weak supervision” comes in a way that makes the agent do so: for example, the weak supervision comes with a low-bias, high-variance noise. In that scenario I think the second term in the objective serves as variance reduction. Q2/A2 for R1 did not resolve my concerns on why PeerRL is better when there is bias in the supervision. I’d appreciate it if the authors can further clarify in that regard.
> >
> >
> > Q2.
> > Thanks for the clarification. I now see how PeerRL + noisy reward might help exploration.
> >
> >
> > Q3.
> > Thanks for the clarification. However, it still concerns me that even with the extension in the appendix, the theorem assumes binary action/rewards or discrete rewards whereas the algorithm operates generically. However, I do think this is a minor point now as the empirical results are strong.
> >
> >
> > Q4.
> > Thanks for the clarification.
> >
> >
> > Q5.
> > Thanks for running the additional experiments.

---

> > > ### Author Response · Authors · 2020-11-24
> > > **Response to AnonReviewer4**
> > >
> > > We are glad that our clarifications addressed your concerns.
> > >
> > > Q1. Thanks for the further clarification. We understand it is not intuitive why the CA objective can correct the biased-noise in the weak supervision perfectly.
> > > First of all, we emphasize that the noise model considered in the paper is actually a form of biased-noise. Therefore, the proposed peer reward is able to handle biased noise in theory.
> > >
> > > Intuitively, both the noisy reward $\tilde{r}$ and peer penalty term $\tilde{r}^\prime$ encodes the noise knowledge implicitly. The carefully constructed form of peer reward allows offsetting the noise ($e_{-}r_{+} + e_{+} r_{-}$, see below equations) in these two terms, leading to a nice property for peer reward - invariant against the noise and is an affine transformation of true reward in expectation.
> > >
> > > To obtain a precise understanding of why the CA objective could offset the noise by penalizing the blind agreement, we have to take a look at the mathematical derivations in Lemma 1.
> > >
> > > We encourage the reviewer to take a look at the proof of Lemma 1. Specifically, we can obtain the expectation of noisy reward from Line 7-13:
> > > $$ \mathbb E [\tilde{r}] = (1 - e_{+} - e_{-})\mathbb E [r] + e_{-}r_{+} + e_{+} r_{-}$$
> > > Similarly, we can get the expectation of peer term from Line 14-16:
> > > $$ \mathbb E [\tilde{r}^\prime] = const + e_{-}r_{+} + e_{+} r_{-},$$
> > > where the const is not dependent on noise rate.
> > > Then we have
> > > $$\mathbb E[\tilde{r}\_\text{peer}] = \mathbb E [\tilde{r}] -  \mathbb E [\tilde{r}^\prime] = (1 - e\_{-} - e\_{+}) \mathbb E[r] + const.$$
> > > The noise residuals $e\_{-}r\_{+} + e\_{+} r\_{-}$ disappears after the subtraction!
> > > As a consequence, we know that the peer reward is invariant to noise $ \mathbb E [\tilde{r}\_\text{peer}] = \mathbb E [r\_\text{peer}]$ and is also an affine transformation of true reward:
> > > $$ \mathbb E [\tilde{r}\_\text{peer}] = (1 - e\_{+} - e\_{-})\mathbb E [r] + const.$$
> > >
> > > As long as $1 - e_{+} - e_{-} > 0$, maximizing the true reward is identical to maximizing the peer reward in expectation.
> > >
> > > Q3: Admittedly, our framework could only generalize to the discrete setting because of the noise model. However, as you recognized, our solution has the potential to deal with the noise in the real world. Moreover, we supplement that in the robust RL/learning in noise literature (Natarajan
> > >  et al., 2016, Wang et al., 2020), the noise model with the confusion matrix is a standard setting to consider. Some other works (Romoff et al, 2018) consider continuous noise but they focus on the variance reduction and assume the noise is zero-mean Gaussian.
> > >
> > > Q5. We have finished the experiments for Pendulum and found that PeerRL could successfully recover from the reward noise. The experimental results are provided in Appendix C.4.
> > >
> > > -----
> > >
> > > [1] Learning with noisy labels. Natarajan et al., 2016.
> > >
> > > [2] Reinforcement learning with perturbed reward. Wang et al., 2020.
> > >
> > > [3] Reward estimation for variance reduction in deep reinforcement learning. Romoff et al., 2018.

---

> > > > ### Comment · AnonReviewer4 · 2020-11-24
> > > > **Further response**
> > > >
> > > > Thanks for the further clarification. It would also be nice to see some analysis on noise model beyond the current binary/discrete one, as the experimental results suggest the algorithm can do so. Your responses did clear my original concerns, and my current rating relfects the updated manuscript.

---

### Official Review · AnonReviewer3 · 2020-10-27
**Good empirical results but the theoretical justification doesn't seem right**

**Rating:** 6
**Confidence:** 3

**Review:**

## Sumary

This paper tackles a very important problem of reinforcement learning/imitation learning in the presence of noisy rewards/labels and uses contemporary literature to motivate a simple solution (in a good sense). The empirical results look encouraging on the benchmark problems but the analysis is a little wanting and doesn't get at the heart of the matter. I think with a little more work this paper can be a very good conference paper but at this stage I can't recommend publication in ICLR.

**Before Author Response** Of course, if the authors address my concerns then I'll increase my rating.
**After Author Response** I think the authors made a good effort to address the concerns and I have recommended to accept the paper.

## Contributions

This paper tackles the problem of learning from imperfect reward/imperfect demonstrations using a new evaluation metric called "Correlated Agreement". The proposed metric "regularizes" learning under weak supervision by penalizing "over-agreement" with the supervision signal. The proposed method is meant to be used in situations where the supervision signal is known to be noisy but it does not estimate parameters of the noisy channel   which is corrupting the supervision signal.

## Strengths

The strongest aspect of the paper in my opinion are the strong empirical results on fairly standard RL benchmarks. The paper also explains the concept of learning from "peer agents". The analysis of the methods presented covers the most obvious questions one might ask at the beginning about the proposed method and the appendices cover those well.

## Weaknesses

There are two problems I see with the analysis in the paper that gives me concern:

1. The main result in the paper is Theorem 1 which relies on Lemma 1. Lemma 1 essentially states that the proposed new metric (Peer RL reward) that subtracts a chance agreement baseline (Second term in equation 1) from the standard agreement objective (First term in equation 1) is an affine function of the true reward. Theorem 1 then crucially relies on this result for the ensuing convergence analysis. The problem I see is that an analogous result to Lemma 1 -- and therefore Theorem 1 -- can also be proved for the noisy reward. Specifically equation (12) in Appendix A.1 shows that even the noisy reward 𝔼[\tilde{r}] is an affine function of the true reward, with the exact same slope and just a different intercept. Given that observation the proof for Theorem 1 can be carried out /mutatis mutandis/ to prove convergence with the naive noisy reward instead of the peer reward. So to me all of the analysis in the current paper doesn't really explain "why" PeerRL is actually performing better than using noisy rewards. To me it seems that more attention must be paid to the intercept terms in equation (12) and (17) to really understand the gap in performance which the authors currently do not consider. This is the first issue I see in the analysis.

2. The second issue I have is with the "Correlated Agreement" Objective itself. The "CA with weak supervision" objective shown in equation (1) seems to give unfair advantage to completely random, but high entropy, weak supervision. For example, in the toy example on Page 4 the authors give a demonstration that the "CA metric" will be 0.375 thereby punishing full agreement with a weak baseline. But if instead the weak baseline was a random and unbiased coin toss then the expected CA objective will be 1 - 0.25 - 0.25 = 0.5 > 0.375. So just because the weak supervision signal had higher entropy and therefore lower chance of random agreement therefore its score was higher. This aspect should be dealt with more thoroughly in the paper.

---

> ### Author Response · Authors · 2020-11-22
> **Response to AnonReviewer3**
>
> We thank the reviewer for the valuable comments on proposed objective function and the theorem.
>
> Q1: the explanation for theorem 1 and the proof
>
> Thanks for pointing it out. Although both the proposed peer reward and the standard objective (noisy reward) are affine functions of the true reward, our proposed peer reward leads to better performance than noisy reward, especially when the data contains more noise. This is because peer reward is invariant to the noise and the constant introduced has the same order of magnitude of reward signals. Therefore, using proposed peer rewards provides stronger true signals in practice.
>
> More specifically, we proved in Line (12) that
> $$
> \mathbb E[\tilde r]
> = (1 - e_- - e_+) \mathbb E[r] + e_- r_+ + e_+ r_-
> = (1 - e_- - e_+) (\mathbb E[r] + \frac{e_-}{1 - e_- - e_+} r_+ + \frac{e_+}{1 - e_- - e_+} r_-),
> $$
> and
> $$
> \mathbb E[\tilde r_\text{peer}]
> = (1 - e_- - e_+) (\mathbb E[r] - (1 - p_\text{peer}) r_- - p_\text{peer} r_+).
> $$
> Since the magnitude of noise terms $\frac{e_-}{1 - e_- - e_+}$ and $\frac{e_+}{1 - e_- - e_+}$ can potentially become much larger than $(1 - p_\text{peer}$ and $p_\text{peer}$ when $e_+$ and $e_-$ are big, $ \frac{e_-}{1 - e_- - e_+} r_+ + \frac{e_+}{1 - e_- - e_+} r_-$ will dilute the informativeness of $\mathbb E[r]$; on the other hand, $\mathbb E[\tilde r_\text{peer}]$ contains less noise than $\mathbb E[\tilde r] $ and thus more information of $\mathbb E[r]$.
>
> For better understanding, we provide a toy example here: assume a binary noisy setting: $e_{-} = e_{+} = 0.45$, $(r_{-}, r_{+}) = (0, 1)$. Then we know that
> $\mathbb E[\tilde r] = 0.2 \mathbb E[r] + 4.5$
> and
> $\mathbb E[\tilde r_\text{peer}] = 0.2( \mathbb E[r] - p_\text{peer}) = 0.2 \mathbb E[r] - 0.2p_{peer} \in (0.2 \mathbb E[r] - 0.2, 0.2 \mathbb E[r] + 0.2)$.
>
> Compared to noisy reward, our approach provides stronger signals such that the agent is able to converge faster in practice (see Figure2). We will add more discussions in the revision.
>
> -------
>
> Q2: Unfair advantage to weak supervision that has high entropy
>
> Instead of stating it as an “unfair” advantage to punish less for the weak supervision sequences that have higher entropy, we think, quite on the contrary, it is necessary to impose less punishment when the agent fully agrees with the high-entropy weak supervision -- perfectly learning from a high-entropy supervision is often a signal of a learning meaningful patterns from a richer set of information. Our algorithm captures this effect precisely - now since there is a lower chance of blind-agreement (“mismatching” a prediction with a highly uncertain supervision for an independent state) under such weak supervision, we would encourage learning of this high-entropy supervision. For instance, assuming we have two weak supervision sequences (1) $a_0 = a_1 = a_2 = … = a_9 = 0$; (2) $a_0 = a_1 = … = a_4 = 0, a_5 = a_6 = … = a_9 = 1$. The second sequence apparently provides more training signals than the first one thus we should impose “less punishment” on the perfect prediction of the second sequence.

---

> > ### Comment · AnonReviewer3 · 2020-11-23
> > **Rejoinder to authors.**
> >
> > ### Q1.
> >
> > > "This is because peer reward is invariant to the noise and the constant introduced has the same order of magnitude of reward signals. Therefore, using proposed peer rewards provides stronger true signals in practice."
> >
> > Thanks for the response. Yes this was also my point that the effectiveness of the peer procedures may be better explained by focusing on these intercept terms rather than the coefficient of 𝔼[r]. But these intercepts do not currently feature in the subsequent theory / convergence proofs. I think they should play a more prominent role.
> >
> > ### Q2.
> >
> > > "it is necessary to impose less punishment when the agent fully agrees with the high-entropy weak supervision"
> >
> > Yes I see it now, when the supervision has high entropy then standard accuracy will be inherently much lower therefore the small reduction in the second term does not hurt us so much.  I agree with you that this is a better way to understand the effect of the peer* objectives. Thanks for the clarification.

---

> > > ### Author Response · Authors · 2020-11-24
> > > **Response to AnonReviewer3**
> > >
> > > We are glad to hear that our clarifications are helpful.
> > >
> > > Q1. focus on the intercept terms of noisy reward and peer reward. How are these terms reflected in the theory?
> > >
> > > Thanks for pointing it out. We agree that these intercepts play a prominent role in practice. We have added a paragraph in the method section to discuss the advantage of proposed peer reward over noisy reward.
> > >
> > > For the convergence and sample complexity analysis, however, this effect cannot be reflected in the theory unfortunately. This is because Q-Learning updates the value function and takes the maximum value as the optimal policy. Such greedy strategy will neglect the noise effect as long it doesn’t flip the order of Q-values at different states. This is similar to the 0-1 loss in the supervised learning, which is also quite robust to noise. To make the noise distinguishable, we assume 1 - e_{-} - e_{+} > 0, which decides that the optimal policy will not change even with noisy reward. However, experiments and the analysis we discussed before indicate that noisy reward may suffer from insufficient training signals thus leading to a significantly worse performance.

---

### Official Review · AnonReviewer1 · 2020-10-28
**This work presents two new methods, PeerRL and PeerBC, which address certain forms of noise in reinforcement learning and behavioral cloning.  Empirical results suggest that these methods are effective at mitigating the effects of noise in rewards and demonstrated actions, but it is unclear that the observed improvements in performance are actually the result of the new methods being more robust to noise.**

**Rating:** 6
**Confidence:** 4

**Review:**

SUMMARY OF CONTRIBUTION:

This work presents a pair of algorithms, PeerRL and PeerBC, which address certain forms of noise in reinforcement learning and behavioral cloning respectively.  For PeerRL, an arbitrary RL algorithm trains on an augmented reward signal, which is generated by subtracting a past reward signal (sampled randomly from the replay buffer) from the current reward.  PeerBC uses an augmented maximum-likelihood behavioral cloning loss that penalizes a policy for log-probability of a randomly sampled action for a randomly sampled state.  Empirical results show that Deuling DQN with the PeerRL reward outperforms the base DDQN algorithm in a noisy-reward version of the cart-pole task.  They also show that PeerBC outperforms standard behavioral cloning in learning from synthetic demonstrations on several Atari games, as well as the cart-pole and Acrobot tasks, all with noisy versions of the base reward signals.  The also evaluate PeerBC in in combination with reinforcement learning in a co-training setup, demonstrating a significant advantage over co-training with standard BC.

AREAS OF CONCERN:

The main concern with this work is that the gains in performance observed with the PeerRL and PeerBC algorithms may not actually reflect the ability of these methods to correct for a specific type of noise.  For PeerRL, the modified reward is fact that PeerRL is able to outperform the version of DDQN with access to the true reward (Figure 2) suggests that at least part of advantage of PeerRL is better scaling of the reward signal, rather than noise reduction.  It would be helpful to compare PeerRL against noiseless scales and shifts of the true reward.  It also seems likely that simply adding a constant baseline value to the noisy reward would help reduce the variance of the return (PeerRL subtracts a noisy baseline value).  Comparing against this simpler variance reduction method would help us understand the importance of the specific form of the augmented PeerRL reward in improving performance.

For PeerBC, the concern is that the modified loss simply biases against high-entropy policies.  When the true policy is itself stochastic, the learned policy found by PeerRL may end up being the mode of this policy.  The advantage of PeerBC observed in Figure 3 may result from the fact that less noisy policies perform better in these tasks.  The fact that PeerBC outperforms the expert policy in Enduro (Figure 3c) would seem to support this hypothesis (the expert's true policy is more stochastic than the learned policy).  It might be helpful to compare PeerBC to a simpler approach which learns a stochastic policy (under the standard maximum likelihood objective), but always takes the most probable action under this policy during evaluation.

While the authors claim to address the case where the rewards or demonstrated actions are perturbed by some arbitrary confusion matrix.  The theoretical and empirical analysis of PeerRL however are limited to the case of binary rewards. Similarly, the theoretical analysis of PeerBC appears to limited to the case of binary actions, though empirical results consider larger action spaces.

More generally, it is not clear that the noise models (for RL and BC) capture the challenges typical of RL and imitation learning (discussed in the first paragraph of the introduction).  In RL, the issue is often not that the reward is noisy in a fixed state, but that it is sparse within the state space, such that the return under a random policy is noisy.  For behavioral cloning, errors in human-generated data often reflects the fact that the human demonstrator has actually selected a suboptimal strategy, rather than the noisy execution of an optimal strategy.  Distributional shift in behavioral cloning does not reflect noise in the training data itself, but instead reflects the compounding error that occurs when we execute an imperfect policy for many timesteps.

These issues with the noise model could be addressed by making it clear earlier in the document exactly what noise models are being considered, and providing examples in the introduction of settings in which such noise would be expected in the rewards or demonstration data.

The authors attempt to describe their approaches to handling noise for RL and BC as instances of a more general peer evaluation algorithm.  The relationship between the RL and BC solutions appears to be superficial, however, as PeerRL and PeerBC seem to address noisy supervision in very different ways.  Furthermore, there are no theoretical results presented that apply to the general algorithm.  The derivation of the RL solution from the common framework (Equation 2) is incomplete, as the loss Eva^RL is never made precise.  It would likely be more clear to the reader if PeerRL and PeerBC were presented as separate, but analogous algorithms for their respective learning problems.

There are also some apparent technical issues with the theoretical results:

1) At no point is it specified that the error rates must be less than 50% for binary rewards or actions. Theorems 1 and 2 are clearly wrong if this condition does not hold.  While and attentive reader should be able to infer this constraint on the noise model, it should be explicitly stated.

2) The r_peer term (the true peer reward) in Lemma 1 is never defined.

3) It is unclear what the variable theta in tau_theta refers to in section 4.2.

4) There seems to be an error in the proof of Lemma 1. The decomposition of the noisy reward in lines (7) and (8) seems to assume that, even in the absence of noise, the expected reward signal will be zero, regardless of the policy being followed, or the current state.  This decomposition should incorporate the probability of true reward given the current state (or averaged over states given the current policy).  This may mean that Lemma 1, and potentially Theorem 1, are incorrect (though it is likely that this can be corrected without substantially changing the contribution of the paper).

CONCLUSION:

Even though the work considers a relatively constrained class of noise models, there are still practical cases where such noise in rewards (and particularly in demonstrated actions) is an issue, and effective methods for dealing with such noise are potentially valuable.  The issue with this work is that it is not yet clear that the proposed methods are actually effective in mitigating these types of noise in RL or BC.  It seems likely that at least part of the improvement in performance stems from other factors, either the preference for more deterministic policies in PeerBC, or the rescaling of the reward signal in PeerRL.  Without resolving these issue, it is impossible to know whether the proposed methods will actually be useful in settings where we have noisy rewards or demonstration data.  There are also technical issues which raise doubts about the validity of the theoretical results presented in this work.

---

> ### Author Response · Authors · 2020-11-22
> **Response to AnonReviewer1 (Part 1)**
>
> We thank the reviewer for thoughtful reviews and comments. The reviewer raises some interesting points that potentially provide more insights into why our solution outperforms other baselines empirically. We address the concerns as follows:
>
> The general response to major concerns:
>
> (1) PeerRL: We agree that there might be some other benefits beyond noise deduction brought up by the proposed framework (as discussed in Sec 5.1): e.g., (1) adding noise and denoising in the early stage encourage the exploration of RL agents (Wang et al., 2020); (2) peer penalty term encourages explorations in RL implicitly; (3) human-specific “true reward” is also imperfect which leads to a weak supervision scenario. However, it is non-trivial to verify these hypotheses and merit future filed studies.
> We thank the reviewer for proposing other possible reasons for this phenomenon. We conducted new experiments as the reviewer suggested and provided the results/analysis in the following. In short, we believe that without the effectiveness of noise correction, the benefits brought by variance reduction and better reward scaling is marginal.
>
> (2) PeerBC: The reviewer’s concern is that the improvements might be brought up by simply biasing against high-entropy policies thus PeerBC is useful when the true policy itself is deterministic. We think this hypothesis is possible since the peer penalty term is high-entropy, thus our CA objective encourages low-entropy (deterministic) policies.
> However, after checking the entropy of fully converged PPO agents, we found that most “expert policies” are not deterministic, which indicates that the major benefits do not come from encouraging deterministic policy. Moreover, for a high entropy policy, the punishment of CA objective will reduce too as the chance of matching a prediction and supervision on two independent tasks are going to be smaller (We would like to refer to Q2 of R3 for more details). Finally, we kindly remind that deterministic reward/policy is a standard setting in robust RL since it is impossible to distinguish the stochasticity and noise without further assumptions.
>
> In summary, we conducted additional experiments as suggested and found that significant improvements cannot be achieved if the proposed approach is not robust to noise. We believe that the verifications of some hypotheses are non-trivial and require more careful and separate efforts.
>
>
> Q1: Comparison between PeerRL agent against the agent with noiseless scaled true reward.
>
> A1: We conduct experiments to compare DDQN algorithms with peer reward or 0.8/0.9 * true reward on CartPole. Here we adopt the constant 0.8 or 0.9 < 1 to mimic the value decrease caused by the peer penalty term. We report the total episode $N_{epi}$ (the lower the better) in the following experiments:
>
> |    \    | r     | $0.8 r$ | $0.9  r$ | $r_{peer}$ |
> |---------|-------|---------|---------|--------|
> | $N_{epi}$ | 101.3 | 97.2    | 97.7    | 89.3   |
>
> As shown in the Table, simple scaling of the reward signal on CartPole will not help the learning of the agent. Furthermore, we also try to scale the noisy reward $\tilde r$ in the noisy setting and have the following results:
>
> |    \    | $0.8\tilde r$ | $0.9 \tilde r$ | $r_{peer}$ |
> |---------|----------------|----------------|--------|
> | e = 0.2 | 117.2          | 121.4          | 85.3   |
> | e = 0.4 | 145.4          | 147.7          | 106.9  |
>
> For reference, the average episode $N_{epi}$ for true reward is 101.2. The agents with scaled noisy rewards still suffer from biased noise and perform significantly worse than PeerRL.
>
>
> Q2: Variance reduction techniques for PeerRL
>
> A2: Thanks for your suggestion. We added the comparisons with the variance reduction technique proposed by previous work (Romoff et al., 2018). However, it doesn’t resolve the challenge when bias presents in the observed rewards. The results are as follows:
>
> |     \         | e = 0.1   | e = 0.2 | e = 0.3 | e = 0.4 |
> |---------------|-----------|---------|---------|---------|
> | noisy         | 114.6     | 123.9   | 133.2   | 163.1   |
> | noisy+VRT     | 111.8     | 118.0   | 129.6   | 157.2   |
> | surrogate     | 110.6     | 113.9   | 115.5   | 125.8   |
> | surrogate+VRT | 112.1     | 112.8   | 112.6   | 119.7   |
> | peer          | 86.2      | 85.3    | 88.8    | 106.9   |
> | peer+VRT      | 95.2      | 86.5    | 93.5    | 117.7   |
>
> Specifically, the variance reduction is slightly helpful for the agent with noisy reward and surrogate reward (Wang et al., 2020) but not as significant as PeerRL. Moreover, it will harm the performance of the proposed peer approach. Hence, we believe that little of the performance improvement of adopting PeerRL comes from variance reduction.

---

> > ### Author Response · Authors · 2020-11-22
> > **Response to AnonReviewer1 (Part 2)**
> >
> > We move our answer to question 3-6 to this part due to the character limit.
> >
> >
> > Q3: Stochastic true policy for PeerBC
> >
> > A3: In theory, our solution could only handle the deterministic true reward or policy. Admittedly, generalizing our solution for more stochastic optimal true policy setting remains a challenging on-going effort. We will clarify this clearly in the revision.
> >
> > However, we kindly remind that deterministic policy/reward is a standard setting in robust RL (Romoff et al., 2018, Wang et al., 2020). This is because It is very challenging to handle stochastic true policy in the robust RL/BC setting. Specifically, it's impossible to distinguish stochasticity and noise without further assumptions. Imagine that the agent chooses action a_0 at state s_0, how can we distinguish if it is the effect of the noise or just the true stochastic true actions.
> >
> > Finally, we check the fully converged PPO agent (a surrogate of perfect expert) on four Atari environments in Table 2 and plot the entropy of learned policies. We found that not all policies are deterministic. Except for Freeway, the entropy of expert policies is always larger than 1. This might also demonstrate the flexibility of our proposed approach in dealing with stochastic true policy in practice.
> > Similarly, we calculate the mean value of the highest action probability over 1000 steps for the full-converged PPO agent:
> >
> > | Boxing | Enduro | Freeway | Pong  |
> > |--------|--------|---------|-------|
> > | 0.579  | 0.664  |  0.903  | 0.492 |
> >
> > This again verifies that the true expert policy we aim to recover might not be fully deterministic, which indicates that the major benefits do not come from encouraging deterministic policy. We will report these results and provide more discussions in Appendix.
> >
> > -----
> >
> > Q4: Binary case for theoretical analysis
> >
> > A4: Our theory analysis is not limited to the binary case (see Remark in Sec 4.2). We provided the theoretical results for multi-outcome extension in Appendix A.3. Moreover, although we make assumptions on the noise model, our empirical results in Sec 5.2 show that our solution has the potential to deal with the noise in the real world (e.g., imperfect expert demonstration in behavior cloning and peer agent’s demonstrations in co-training).
> >
> > -----
> >
> > Q5: Issues in the theoretical analysis
> >
> > A5: Thanks for pointing them out. We will clarify them in the revision.
> >
> > (1) Our lemmas and theorems do require this standard assumption $e_{-} + e_{+} < 1$ in noisy learning as shown in the proofs.
> >
> > (2) We will add the definition of r_peer is defined as follows:
> > $r_{\mathrm{peer}}(s, a) = r(s, a) - \xi \cdot r^\prime$
> >
> > (3) $\tilde\tau_\theta$ is the trajectory generated by $\pi_\theta$ with the noisy reward function $\tilde r$.
> >
> > -----
> >
> > Q6: Why not condition on the current state or policy when decomposing the noisy reward?
> >
> > A6: The expectation of the noisy reward is taken over the state visitation measure induced by the policy. The decomposition in Line (6) and (7) does not involve the expansion of probability of true reward given the states, which is covered in the expectation $\mathbb E_{r = r_{-}}$ and $\mathbb E_{r = r_{+}}$. The relation between the expectation of noisy reward and true reward holds for any policies.
> >
> > ------
> >
> > [1] Reward estimation for variance reduction in deep reinforcement learning. Romoff et al., 2018.
> >
> > [2] Reinforcement learning with perturbed reward. Wang et al., 2020.

---

> > > ### Comment · AnonReviewer1 · 2020-11-23
> > > **Response to rebuttals**
> > >
> > >
> > > I appreciate the authors' hard work in addressing my concerns, and have updated my score to reflect the latest version of the paper.  I still have some concerns, however:
> > >
> > > The additional experiments with PeerRL seem to confim my suspicions that much of the benefit of the approach is due to something other than its ability to correct for noisy rewards.  The fact that PeerRL performs significantly better than an agent learning from the true, noiseless reward suggests that, instead of or in addition to the potential noise reduction, PeerRL makes the true reward signal easier to learn from.  This does not mean that the PeerRL reward is not useful, and the thorough comparisons against other variance reduction techniques suggest that it may be particularly effective when dealing with noisy rewards.  It is important however to highlight that there may be multiple explanations for the improvement in performance provided by PeerRL, and which explanation is most relevant will depend on the particular RL task.
> > >
> > > Similarly, in the evaluation of PeerBC, it should be noted that the "expert" policy learned by PPO is not actually the optimal policy in terms of expected return.  This is not surprising, as PPO penalizes low-entropy policies.  The entropy estimates of the expert PPO policies actually seem to confirm my original suspicion. The noise in the exper policy causes it to be suboptimal, and a method which corrects for this noise (even if the noise is relatively small) can actually learn a superior policy to the expert's.  If the expert's policy were truly corrupted by noise (as opposed to simply being stochastic), then PeerBC would seem to be able to compensate for this noise.  In the experiments presented here, the benefits of PeerBC seem to come primarily from its ability to correct for the limitation of the expert's policy
> > >
> > > This is not to say that the PeerBC loss is not advantageous, it is just important to discuss the potential sources of its advantage, and the conditions under which it may perform poorly.  For example, there are settings in which our goal is to match human behavior in distribution, so as to make the behavior of the agent appear more human like.  In such a setting, we may not want to use PeerBC, but if our goal is simply to maximize task reward, then PeerBC may be a good choice.
> > >
> > > I believe there is still an error in the proof of Lemma 1, though the Lemma itself is correct.  Consider the case where, under the current policy, the reward is always r- = 0 (r+ = 1), and where e- = 0, e+ = 0.4,  here, the expectation of the noisy reward is 0 (because the true reward is 0 and there is no error when the reward is zero because e- = 0).  Evaluating equations 7-10, we find that they yield a value of 0.6, and so do not hold.  The step to equation 11 cancels out the original error of not wieghting the expectations by the policy-dependent reward probabilities, and equation 11 holds true.  The proof can be corrected without changing the result.

---

> > > > ### Author Response · Authors · 2020-11-24
> > > > **Response to AnonReviewer1**
> > > >
> > > > We thank the reviewer for the insightful comments and in-depth analysis.
> > > >
> > > > We agree that the benefits of  PeerRL and PeerBC might partially come from either the wise reward scaling or biasing towards low-entropy policies, which are non-trivial observations and worth further exploration in the future. We add two paragraphs in the experiment section to discuss the potential beneficial factors that our approach introduces.
> > > >
> > > > For the PeerBC evaluations, we understand that even if the fully converged PPO agents are ideally treated as the upper bound (a surrogate expert), it may not reflect the true optimal policy. To further verify this, we will conduct additional experiments to check the entropy of policy trained with PeerBC and see whether the uncertainty decreases a lot. If it is true, this also raises another interesting question for future work: whether we can further improve the fully converged PPO using PeerBC strategy.
> > > >
> > > > It's also very interesting that there are some scenarios where PeerBC might not be preferred. For instance, if we aim to make the behavior of the agent appear more human-like, then the ''inherent noise' that leads to suboptimal policy' may be necessary and we don’t want to correct it.
> > > >
> > > > We also thank the reviewer for explaining the notation issues in Lemma 1 clearly. We have fixed the problem in the updated manuscript. Specifically, we weigh the two terms with the probability of getting each reward given the current policy.

---

### Official Review · AnonReviewer2 · 2020-11-01
**Interesting work on learning from noisy supervisions**

**Rating:** 6
**Confidence:** 3

**Review:**

This paper formulates a framework for reinforcement learning and behavior cloning from weak supervisions (i.e., noisy rewards or imperfect expert demonstration). Specifically, it proposes PeerPL to perform efficient policy learning from the available weak supervisions, which covers PeerRL (for RL with noisy rewards), PeerBC (for imitation learning from imperfect demonstration) and PeerCT (for hybrid setting). The PeerPL idea is based on a new weak supervision objective that is in the form of difference between norm learning loss and a loss incurred by randomly sampling the supervision signals. Experimental results demonstrate that PeerPL significantly outperforms SOTA solutions when the complexity or the noise of the learning environment grows. The proposed idea is useful in practice as it increases the robustness of the learning process to imperfect supervision signals.

Comments:
1.	The proposed idea seems to share some similar spirit as the contrastive learning, where it also constructs a contrastive training loss from positive and negative examples. The CA loss (1) is somehow similar to contrastive learning with one negative sample. It would be helpful for the authors to add some further discussion in this regard.

2.	In the illustrative toy example, it would be helpful to explain how the numbers 0.75 and 0.25 come up. Based on my understanding, it seems that the calculation should be 1 – (0.8^2+0.2^2)=0.32?  (The probability of a_j and a_k’ being 1 is 4/5 and the probability of them being 0 is 1/5. So the probability of them being equal is (4/5)^2 + (1/5)^2.) Also, it is necessary to clarify that Eva is chosen to be the indicator loss here. Otherwise, it could be confused with the fact that cross-entropy loss is used in practical implementation.

3.	Based on Lemma 1, in order to guarantee convergence in Theorem 1, shouldn’t there be an additional condition of: e_{+} + e_{-1} < 1 (i.e., not too noisy) in Theorem 1 (to keep the same sign of the reward)? Likewise, I think there should be a similar requirement on the PeerBC setting as well (as reflected in the 1-e_{-}+e_{+} in the denominator of the bound in Theorem 2). If this is true, I think this point should be clarified as we do need some (weak) prerequisite/knowledge about noisy model. If this is not required, then it would be important to show some further experiments on this extremely noisy setting of e_{-}+e_{+} > 1. (Note that currently it only shows up to e=0.4.) In addition, I think it would be better to put all other assumptions (if any) on the noise model in the Theorem.

4.	Regarding the sample complexity result of PeerRL relative to RL with true rewards below Theorem 1, I think it might be useful to put it as a separate theorem after Theorem 1.

5.	Please also report the perfect expert (fully converged PPO) results in Table 2 as a performance upper bound.

---

> ### Author Response · Authors · 2020-11-22
> **Response to AnonReviewer2**
>
> Thanks for your valuable suggestions and positive comments. We will revise the paper accordingly as suggested.
>
> Q1: Discussion about contrastive learning
>
> A1: Thanks for the suggestion. We will add the discussion with contrastive learning in the revision. In general, the proposed CA objective shares a similar spirit as contrastive loss - “encouraging positive examples and penalizing negative examples”. We agree that the CA loss is similar to contrastive learning with one negative sample. Here, the negative example is a mislabeled (precisely random-labeled) example. However, it is worthwhile to notice CA loss and contrastive loss are dealing with different problems (weak supervision in policy learning vs unsupervised learning).
>
> Q2: Clarification on the toy example
>
> A2: Thanks for pointing it out. We made a typo in this example. The sequence of actions should be
> $a_1 = a_2 = a_3 = 1, a_4 = 0$. Otherwise, your calculation is correct. We will also clarify that Eva here is chosen to be the indicator loss as you suggested.
>
> Q3: Clarifications of the assumption $e_{-} + e_{+} < 1$ in lemmas/theorems
>
> A3: Our lemmas and theorems do require this standard assumption $e_{-} + e_{+} < 1$ in noisy learning as shown in the proofs. We will clarify this in the revision.
>
> Q4: Sample complexity should be a separate theorem
>
> A4: We agree that the analysis of sample complexity is one of the contributions in this paper that should be highlighted, currently the analysis of sample complexity is provided in the Appendix with a separate theorem with proof. We will move it to the main paper.
>
> Q5: Report the perfect expert (fully converged PPO) results in Table 2 as a performance upper bound.
>
> A5: Thanks for the suggestion, we conduct additional experiments to train the perfect expert (fully converged PPO):
>
> |  Env   | Score         |
> |---------|:-------------:|
> | Boxing  | 89.3 ± 5.4  |
> | Enduro  | 389.6 ± 216.9|
> | Freeway | 33.3 ± 0.8    |
> | Pong    | 20.9 ± 0.3    |
>
> We will report those results as an upper bound in Table 2 and Figure 3.

---

### Author Response · Authors · 2020-11-24
**Revised manuscript has been uploaded**

We would like to thank the reviewers again for their thoughtful reviews and valuable comments. We have improved the paper according to the comments/suggestions. The changes of manuscript are highlighted with the blue color.

The key changes include

(1) Add additional experiments and analysis to explore multiple possible factors that brings up benefits for PeerRL/PeerBC -> (a) reward scaling (b) variance reduction (Appendix C.6), (c) stochastic true policy for PeerBC (Appendix C.7)

(2) Add more discussions on why proposed peer reward provides privileges over noisy reward.

(3) Add the experiment of PeerRL on one continuous control task - Pendulum (Appendix C.4).

(4) Motivate the noise model in policy learning more properly (Sec 1)

(5) Fix the missing definitions, typos and notation errors in theorems/proofs.

(6) Report the performance of the fully converged PPO agent in Table 2 as an upper bound.

---

### Decision · Program_Chairs · 2021-01-07
**Final Decision**

**Decision:**

Reject

**Comment:**

The reviewers had some initial concerns about this submission. While the authors' rebuttal does a good job to address these concerns, the reviewers still have some doubts about the contribution of this paper and potential impact. In particular, it is not clear whether the performance improvements observed with the proposed algorithms is due to the ability to correct for noisy rewards or whether there are multiple other explanations for the improvement in performance. This makes it hard to predict whether the proposed algorithms will actually be useful in settings where noisy rewards or demonstration data are present.